# Anomaly and invertible field theory with higher-form symmetry: Extended group cohomology

**Shi Chen**

*School of Physics and Astronomy, University of Minnesota, Minneapolis, MN 55455, USA*

*E-mail:* chen8743@umn.edu

ABSTRACT: In the realm of invertible symmetry, the topological approach based on classifying spaces dominates the classification of 't Hooft anomalies and symmetry protected topological phases. We explore the alternative algebraic approach based on cochains that directly characterize the lattice lagrangian of invertible field theories and the anomalous phase factor of topological operator rearrangements. In the current literature, the algebraic approach has been systematically described for only finite 0-form symmetries. In this initial work, we generalize it to finite higher-form symmetries with trivial higher-group structure. We carefully analyze the algebraic cochains and abstract a purely algebraic structure that naturally generalizes group cohomology. Using techniques from simplicial homotopy theory, we show its isomorphism to the cohomology of classifying spaces. The proof is based on an explicit construction of Eilenberg-MacLane spaces and their products.

# 1 Introduction

Symmetry plays a salient role in nonperturbative dynamics. It is believed that in the quantum/statistical world, anything wild can happen unless a symmetry controls it. Today, the connotation of symmetry has become very broad, profound, and sophisticated.

**Generalized invertible symmetry**

Although spacetime symmetry has been generalized long ago (conformal symmetry, supersymmetry, etc.), internal symmetry has been generalized only recently. The starting point is to realize that a traditional symmetry operation is realized by an *invertible codimension-1 topological operator*. Generalization just means to give up the restrictive attributives in the front of "topological operator". Originating from the study of topological line defects in rational conformal field theories (e.g. [1]), abandoning "invertible" has been drawing prominent attention in recent few years. Nevertheless, we exclusively focus on invertible symmetries throughout this work.

Abandoning "codimension-1" was largely motivated by the study of line operators in gauge theories [2–5] and topological field theories [6]. In the seminal work [7], the symmetry generated by codimension-$(p+1)$ topological operators is called a $p$-form symmetry. Higher-form symmetries have to be commutative due to the obvious dimensional reason. Let us present two of the most famous slogans about higher-form symmetries: Photons are Nambu-Goldstone bosons of (probably emergent) 1-form U(1) symmetries; topological orders are spontaneous breaking of (typically emergent and probably non-invertible) discrete higher-form symmetries.

Around the time of Ref. [7], it was also noticed that higher-codimensional topological operators a priori live on the junctions between lower-codimensional topological operators, and the admissible connection means of higher-codimensional topological operators are a priori influenced by the ambient lower-dimensional topological operators [8, 9]. Mathematically, this comes from an extension of lower-form symmetries by higher-form symmetries, in many aspects much akin to an ordinary group extension. A web of connected topological operators gives as a concrete realization of a background gauge field (especially for discrete symmetry), and the rearrangements of junctions give gauge transformations. Hence given the same spectrum of $p$-form symmetries, different extensions require different types of gauge fields and thus mean different symmetries. These distinct algebraic structures are called higher-groups. We claim that *the most generalized invertible symmetries are described by higher-groups.*

The most tractable way to characterize a higher-group is to describe its classifying space. In this topological approach, extensions are realized by fibrations. The classifying space of a symmetry classifies the gauge fields of the symmetry. More precisely, the topologically inequivalent gauge fields on a topological space $X$ are classified by the homotopy classes of maps from $X$ to the classifying space. Hence classifying spaces are well-defined only up to homotopy equivalence. A discrete higher-group is completely determined by the homotopy type of its classifying spaces. Conversely, any topological space is a classifying space of a discrete higher-group.

Let us describe extensions slightly more concretely. For a $p$-form symmetry $G$, Abelian when $p > 0$, its classifying space is its $(p+1)$-th iterated delooping $B^p G$. For a 2-group comprised by a 0-form symmetry $G_1$ and a 1-form symmetry $G_2$, its classifying space $X$ fits into a fibration $B^2 G_2 \to X \to BG_1$. For a 3-group that further takes a 2-form symmetry $G_3$ into account, its classifying space $Y$ fits into a fibration $B^3 G_3 \to Y \to X$. Iterating this procedure form by form, we can obtain the classifying space of an $n$-group for arbitrary $n$. An $n$-group is thus completely characterized by the $n-1$ fibrations, in addition to the spectrum of $p$-form symmetries $G_1, G_2, \cdots, G_n$. When the symmetry is discrete, the above procedure is called a Postnikov tower.

## Anomaly inflow and invertible field theory

An invertible symmetry gets extremely powerful if it has an 't Hooft anomaly. An anomaly is an obstruction to coupling the symmetry with a background gauge field: No local counterterm can remedy gauge invariance and the phase of the partition function is ambiguous under gauge transformations. Anomalies are powerful because they are preserved by renormalization group flow and thus constrain the infrared behavior of a quantum/statistical system: The existence of an anomaly strictly excludes a trivially gapped phase. An anomalous discrete symmetry still allow gapped phases via spontaneous breaking, where the infrared is a nontrivial topological field theory. Apart from this possibility, the gap has to be closed by massless degrees of freedom such as conformal, Nambu-Goldstone, or chiral-fermionic modes. Traditional examples include the Lieb-Schultz-Mattis theorems [10–16] in spin systems and chiral symmetry breaking in fermionic gauge theories [17–23]. Conversely, as long as the theory is not trivially gapped, in the infrared necessarily emerge anomalous symmetries, which might persist into ultraviolet or might not. In particular, in spontaneous breaking of non-anomalous symmetry always emerges an extra symmetry that has a mixed anomaly with the original symmetry.

We may be satisfied with a covariant partition function under gauge transformations but we can actually remedy gauge invariance by a local counterterm one dimension higher. More accurately, we regard an $n$-dimensional anomalous theory as living on the boundary of an $(n+1)$-dimensional invertible field theory with the same symmetry. A field theory is said invertible if it always has 1-dimensional Hilbert spaces. Its partition is thus always a phase factor on any closed spacetime and loses gauge invariance if the compact spacetime has a boundary. If a properly anomalous theory lives on the boundary, the whole system restores gauge invariance. This is called anomaly inflow and establishes a bijective correspondence between $n$-dimensional anomalies and $(n+1)$-dimensional invertible field theories. It is

thus convenient to characterize an anomaly by its associated invertible field theory through anomaly inflow.

Invertible field theories themselves also describe physical systems. In the presence of a symmetry, trivially gapped phases can be further differentiated into distinct symmetry protected topological phases [24]. Typical examples include the Haldane phase [25–27], the Kitaev $E_8$ phase [28–30], the Gu-Levin $\mathbb{Z}_8$ phases [31, 32], negatively massive fermions [33–36], and $2\pi$-differed $\theta$-angles in gauge theories [37, 38]. The difference between two symmetry protected topological phases is exactly characterized by an invertible field theory [39–41]. Namely, the infrared of one phase is the infrared of another phase tensored with an invertible field theory. Depending on the explicit physical system, there might be a preferred trivial phase (the former four examples above) or no preferred trivial phase (the last example above). In general, symmetry protected topological phases form a principal homogeneous space under the group of invertible field theories[1]. Due to anomaly inflow, an interface between two symmetry protected topological phases either explicitly violates the symmetry or renders the symmetry anomalous. Hence a symmetric interface cannot be trivially gapped and must have nontrivial infrared dynamics.

**Topological approach vs. algebraic approach**

The characterization and classification of invertible field theories are of the first priority in the study of 't Hooft anomalies and symmetry protected topological phases. Based on the analysis of partition functions, invertible field theories are argued to be characterized and classified by an appropriate cohomology theory related to the classifying space [29, 43]. The simplest Ansatz is the integral cohomology [44]. More precisely, $\bullet$-dimensional invertible field theories are classified by $H^{\bullet+1}(X, \mathbb{Z})$ for the classifying space $X$. When the symmetry is finite (i.e., discrete and compact), the integral cohomology is isomorphic to the shifted U(1) cohomology, i.e., $H^{\bullet+1}(X, \mathbb{Z}) = H^{\bullet}(X, U(1))$. Such ordinary cohomologies suffice for simple bosonic systems.

The pursuit of more complicated systems involving fermions, time-reversal symmetry, gravitational anomalies, etc., requests more sophisticated generalized cohomologies [29, 43, 45]. In different circumstances apply different cohomologies. Today, it is widely believed that bordism cohomologies provide universal solutions [39, 46–50]. Here a bordism cohomology means the generalized cohomology given by the shifted Anderson dual of an appropriate Thom spectrum. The choice of the Thom spectrum is determined by the symmetry details of the physical systems and is constructed out of the classifying space. A piece of strong evidence comes from the axiomatic treatment of invertible fully-extended topological field theories [39, 50]. Another rationale comes from the study of fermionic path integral [33–36], where invertible field theories can be systematically captured by the Atiyah-Patodi-Singer $\eta$-invariant through the Dai-Freed theorem [51]. The $\eta$-invariant is a bordism invariant for (many different) appropriate manifold classes.

---

[1]This relative nature of symmetry protected topological phases is recently emphasized by the authors of Ref. [42]. They observed that this relative nature becomes particularly significant if the symmetry ceases to be invertible.

We refer to the above treatment as the topological approach, which is primarily based on the partition function of invertible field theories. By this paper, we initiate an exploration on the alternative algebraic approach that is instead based on the Dijkgraaf-Witten-type discrete formulation of invertible field theories. In the algebraic approach, anomaly inflow is pretty intuitive and directly characterizes the anomalous phase factor accompanying a discrete gauge transformation, i.e., a junction rearrangement in a web of connected topological operators. The topological and the algebraic approaches *are supposed to* be equivalent to each other. They ought to yield the same classification of invertible field theories, as well as the same classification of 't Hooft anomalies and symmetry protected topological phases. They should be reciprocal and complementary to each other such that we can freely switch from one to the other from time to time.

The ideal vision above is far from the current reality. The algebraic approach was systematically studied for finite 0-form symmetries and a few cohomologies only. Group cohomology [52] describes bosonic systems and is isomorphic to the ordinary cohomology of classifying spaces. Group supercohomologies, including the original version [53] and an updated version [54–56], describe fermionic systems and are isomorphic to certain generalized cohomologies of classifying spaces that approximate the spin bordism cohomology from low degrees [45]. No systematic algebraic approach beyond these has been established to adapt higher-form symmetries, bordism cohomologies, or continuous symmetries. Although pretty impressive progresses have been made to understand the anomalies and invertible field theories of 2-groups [8, 57, 58], we are still quite far away from a satisfactory algebraic approach for general invertible symmetries that is equivalent to the topological approach. This paper is devoted to a step forward toward this direction.

## 1.1 Convention and summary

To make our narrative smooth, we introduce the following convenient notion which we shall extensively use throughout this paper.

**Definition 1.1.** An *extended group* $G$ is a sequence of groups $G_1, G_2, G_3, \cdots$ such that (i) $G_k$ is Abelian for all $k > 1$; (ii) there is a number $N$ such that $G_k$ is trivial for all $k > N$; (iii) every $G_k$ is at most countable.

Given that $G_1$ can be non-Abelian, we describe the group composition of $G_k$ using the multiplicative convention. Their identities are denoted by 1. An extended group $G$ exactly describes the algebraic structure of a discrete invertible symmetry with trivial higher-group structure. Namely, $G_{p+1}$ describes the $p$-form discrete invertible symmetry while the absence of the extension information means trivial higher-group structure. Hence we shall abbreviate a "discrete invertible symmetry with trivial higher-group structure" as an "extended-group symmetry".

Our goal in this work is to explore the elementary cochain description of anomalies and invertible field theories for finite extended-group symmetries. The physical results will usually apply to finite extended groups only while the mathematical results will apply to general extended groups. The paper is organized as follows.

**(I)**  In Sec. 2, we shall construct *extended group cohomology* that characterizes the lattice lagrangian of invertible field theories.

**(II)**  In Sec. 3, we shall show the isomorphism between extended group cohomology and ordinary cohomology of classifying spaces.

**(III)**  In Sec. 4, we shall describe 't Hooft anomalies using the algebraic cochains of extended group cohomology.

## 2 Invertible field theory: Algebraic approach

We investigate the discrete formulation of invertible field theories with finite extended-group symmetries. In Sec. 2.1, we count the degrees of freedom and characterize the lattice lagrangian of invertible field theories. In Sec. 2.2, we isolate the purely algebraic structures from the geometric background and mathematically formulate the resulting cohomology. In Sec. 2.3, we elaborate on the low-dimensional cases with illustrations.

### 2.1 Discrete formulation

Let us consider an $n$-dimensional invertible field theory with a discrete symmetry described by a finite extended group $G$. The compactness of the symmetry ensures the unitarity of the invertible field theory. Its partition function on a closed smooth $n$-manifold $\mathcal{M}$ is given by

$$\mathrm{e}^{\mathrm{i}\mathcal{S}}, \qquad \mathcal{S} \sim \mathcal{S} + 2\pi. \tag{2.1}$$

The action $\mathcal{S}$ is a $U(1)$-valued function on the space of all background $G$ gauge fields. A $G$ gauge field contains a $k$-form $G_k$ gauge field for each $k$, which is necessarily locally flat and described by a cohomology class $\in H^k(\mathcal{M}, G_k)$.

Let us formulate invertible field theories on a discrete spacetime. Following Ref. [59], we triangulate[2] the spacetime manifold and discretize the $G$ gauge field. Namely, we assign a $G_1$ element to each 1-simplex, a $G_2$ element to each 2-simplex, and so on. Since the gauge field has to be locally flat, the $G$ element on each simplex is not independent. We are going to carefully inspect the independent degrees of freedom in a single $n$-simplex.

#### 2.1.1 Independent degrees of freedom

Let us first introduce more accurate language to describe the gauge fields on an $n$-simplex. An $n$-simplex is spaned by $n+1$ vertices and we label them by $0, 1, \cdots, n$. Then a $k$-facet is uniquely specified by $k+1$ vertices. Let us encode this in the following notation.

**Definition 2.1.** For non-negative integers $n$ and $k \leq n$, we define $\langle n : k \rangle$ to be the set of all $(k+1)$-element subsets of $\{0, 1, \cdots, n\}$.

$\langle n : k \rangle$ represents the set of all $k$-facets in an $n$-simplex. When we refer to a $k$-facet $\{\ell_0, \ell_1, \cdots, \ell_k\} \in \langle n : k \rangle$, we always take the vertices in the increasing order $0 \leq \ell_0 < \ell_1 < \cdots < \ell_k \leq n$. We call this the standard orientation. When we do a sum/product over the boundary $(k-1)$-facets of a $k$-facet, we should instead take the Stokes-law orientation with respect to the $k$-facet. The Stokes-law orientation of the $(k-1)$-facet $\{\ell_0, \ell_1, \cdots, \ell_k\} - \{\ell_j\}$ with respect to the $k$-facet $\{\ell_0, \ell_1, \cdots, \ell_k\}$ differs from the standard orientation by the sign

$$(-)^j. \tag{2.2}$$

It comes from the fact that popping $\ell_j$ to the beginning of $\{\ell_0, \ell_1, \cdots, \ell_k\}$ needs $j$ times of switching an adjacent pair.

---

[2]More accurately, by triangulation we mean the more flexible $\Delta$-complexes (see [60, Sec. 2.1]), which generalize simplicial complexes. Every smooth manifold is triangulizable.

The total number of $k$-facets in an $n$-simplex is given by the cardinality of $\langle n\!:\!k \rangle$,

$$\frac{(n+1)!}{(k+1)!(n-k)!} \, . \tag{2.3}$$

On an $n$-simplex, a $G$ gauge field, comprised of $G_k$ gauge fields, is specified by maps

$$A : \langle n\!:\!k \rangle \mapsto G_k \tag{2.4}$$

for each $k$. $A$ is strictly constrained by the locally-flatness condition. Accurately speaking, we call a $G_k$ gauge field locally flat if the product of the $G_k$ gauge field over the boundary of any $(k+1)$-facet (under the Stokes-law orientation) is trivial. Namely, $A$ is locally flat if

$$\prod_{j=0}^{k+1} A_{\ell_0, \cdots, \ell_{j-1}, \ell_{j+1}, \cdots, \ell_{k+1}}^{(-1)^j} = 1 \, , \qquad \forall k, \ \forall \{\ell_0, \ell_1, \cdots, \ell_{k+1}\} \in \langle n\!:\!k\!+\!1 \rangle \, . \tag{2.5}$$

The constraints themselves are also not independent. It is not hard to notice that the gauge fields on all the facets that contain a fixed vertex, say 0, constitute a set of independent degrees of freedom. Hence the number of independent degrees of freedom is

$$\frac{n!}{k!(n-k)!} \tag{2.6}$$

for each $k$. However, we shall not use this set of independent degrees of freedom. Instead, we shall take another ansatz as follows. Let us first introduce a new notion.

**Definition 2.2.** For positive integers $n$ and $k \leq n$, we define $[n\!:\!k]$ to be the set of all $k$-element subgroups of $\{1, 2, \cdots, n\}$.

$[n\!:\!k]$ has no direct geometric meaning while its cardinality is exactly the number (2.6). When we refer to $\{q_1, \cdots, q_k\} \in [n\!:\!k]$, we still stick to the increasing order $1 \leq q_1 < \cdots < q_k \leq n$. We now encode the independent degrees of freedom of $A$ in the maps

$$g : [n\!:\!k] \mapsto G_k \tag{2.7}$$

through the ansatz

$$A_{\ell_0, \ell_1, \cdots, \ell_k} = \prod_{q_1=\ell_0+1}^{\ell_1} \prod_{q_2=\ell_1+1}^{\ell_2} \cdots \prod_{q_k=\ell_{k-1}+1}^{\ell_k} g_{q_1, q_2, \cdots, q_k} \, , \tag{2.8}$$

which, we verify, ensures the locally-flatness condition (2.5). Through this Ansatz, we have expressed $A_{\ell_0, \ell_1, \cdots, \ell_k}$ as the product of

$$(\ell_1 - \ell_0)(\ell_2 - \ell_1) \cdots (\ell_k - \ell_{k-1}) \tag{2.9}$$

independent degrees of freedom. $A_{\ell_0, \ell_1, \cdots, \ell_k}$ itself is an independent degree of freedom if and only if $\ell_0, \ell_1, \cdots, \ell_k$ are consecutive integers and then we have $A_{\ell_0, \ell_1, \cdots, \ell_k} = g_{\ell_1, \cdots, \ell_k}$.

### 2.1.2   Lattice lagrangian

In the discrete formulation, an invertible field theory is specified by a lattice lagrangian,

$$\mathcal{L} \sim \mathcal{L} + 2\pi \, . \tag{2.10}$$

It is a $U(1)$-valued function on the space of locally flat $G$ gauge fields on an $n$-simplex. The sum of a lagrangian over all $n$-simplices in the triangulation gives an action, i.e.,

$$\mathcal{S} \; = \sum_{\text{all } n\text{-simplices}} \mathcal{L} \, . \tag{2.11}$$

Having extracted the independent degrees of freedom in an $n$-simplex, we regard a lattice lagrangian as a function of $g$ instead of $A$.

Crucially, an action $\mathcal{S}$ should not depend on the choice of a particular triangulation. This is the discrete version of gauge invariance and strictly constrains lagrangians $\mathcal{L}$. Let us consider the elementary subdivision that divides a single $n$-simplex into $n{+}1$ $n$-simplices. Putting it in a fancier way, we attach an $(n{+}1)$-simplex to this $n$-simplex and replaces it by other $n$-facets of the $(n{+}1)$-simplex. Hence the condition of gauge invariance can be stated as follows: The sum of an $n$-dimensional lagrangian over the boundary of an $(n{+}1)$-simplex (under the Stokes-law orientation) should be trivial.

In addition to gauge invariance, there are also gauge redundancies. Some nontrivial lagrangians $\mathcal{L}$ always sum up to a trivial action. Such an $n$-dimensional lagrangian on an $n$-simplex is exactly given by the sum of an $(n{-}1)$-dimensional lagrangian over the boundary of this $n$-simplex (under the Stokes-law orientation). We regard these $n$-dimensional lagrangians as gauge redundancies and modulo them out when we characterize and classify lagrangians.

In both gauge invariance and gauge redundancy appears a common operation, i.e., obtaining an $n$-dimensional lagrangian via summing an $(n{-}1)$-dimensional lagrangian over the boundary of an $n$-simplex (under the Stokes-law orientation). This boundary-sum operation provides a homomorphism from the group of $(n{-}1)$-dimensional lagrangians to the group of $n$-dimensional lagrangians. Let us denote it by $d_{n-1}$. Then the equivalence classes of legitimate $n$-dimensional lagrangians can be compactly expressed by

$$\frac{\ker d_n}{\operatorname{im} d_{n-1}} \, . \tag{2.12}$$

This is nothing but the cohomology of a cochain complex made by lagrangians and $d_n$'s. Our goal is thus to find out the accurate expression for $d_n$.

For this purpose, let us embed an $(n{-}1)$-simplex into an $n$-simplex as one of its $(n{-}1)$-facets. The embedding to the $(n{-}1)$-facet that does not contain the vertex $j$ is abstractly specified by an order-preserving injective map between the sets of vertices, i.e.,

$$\eta_j \, : \, \{0, 1, \cdots, n{-}1\} \mapsto \{0, 1, \cdots, n\} \, , \qquad j = 0, 1, \cdots, n \tag{2.13}$$

where $j$ indicates the vertex that falls off the image of $\eta_j$. Explicitly,

$$\eta_j(\ell) \equiv \begin{cases} \ell \, , & \ell < j \\ \ell + 1 \, , & \ell \geq j \end{cases} \, . \tag{2.14}$$

Accurately speaking, our goal is to find out $\eta_j^* g$, the pullback of $g$ under the embedding $\eta_j$. First, we can readily find the pullback of $A$, (with $0 \leq \ell_0 < \cdots < \ell_k \leq n-1$)

$$
\begin{aligned}
\left(\eta_j^* A\right)_{\ell_0, \ell_1, \cdots, \ell_k} &\equiv A_{\eta_j(\ell_0), \eta_j(\ell_1), \cdots, \eta_j(\ell_k)} \\
&= A_{\ell_0, \cdots, \ell_{r-1}, \ell_r+1, \cdots, \ell_k+1}, \qquad \ell_{r-1} < j \leq \ell_r.
\end{aligned}
\tag{2.15}
$$

Plugging this into Ansatz (2.8), we see that the upper bound of the $r$-th product changes from $\ell_r$ to $\ell_r + 1$, and all the subsequent products simultaneously shift the lower and upper bounds by $+1$. Explicitly, we have

$$
\left(\eta_j^* A\right)_{\ell_0, \ell_1, \cdots, \ell_k} = \prod_{q_1=\ell_0+1}^{\ell_1} \cdots \prod_{q_{r-1}=\ell_{r-2}+1}^{\ell_{r-1}} \prod_{q_r=\ell_{r-1}+1}^{\ell_r+1} \prod_{q_{r+1}=\ell_r+2}^{\ell_{r+1}+1} \cdots \prod_{q_k=\ell_{k-1}+2}^{\ell_k+1} g_{q_1, \cdots, q_k}, \quad (2.16)
$$

with $\ell_{r-1} < j \leq \ell_r$. Comparing it with the original Ansatz (2.8) for an $(n-1)$-simplex, i.e.,

$$
\left(\eta_j^* A\right)_{\ell_0, \ell_1, \cdots, \ell_k} = \prod_{q_1=\ell_0+1}^{\ell_1} \prod_{q_2=\ell_1+1}^{\ell_2} \cdots \prod_{q_k=\ell_{k-1}+1}^{\ell_k} \left(\eta_j^* g\right)_{q_1, q_2, \cdots, q_k}, \tag{2.17}
$$

we can extract (with $1 \leq q_1 < \cdots < q_k \leq n-1$)

$$
\left(\eta_j^* g\right)_{q_1, q_2, \cdots, q_k} = \begin{cases} g_{q_1, \cdots, q_\bullet, q_{\bullet+1}+1, \cdots, q_k+1}, & q_\bullet < j < q_{\bullet+1} \\ g_{q_1, \cdots, q_\bullet, q_{\bullet+1}+1, \cdots, q_k+1} \, g_{q_1, \cdots, q_\bullet+1, q_{\bullet+1}+1, \cdots, q_k+1}, & q_\bullet = j \end{cases}. \tag{2.18}
$$

Here the extra multiplicand on the second line accommodates the additional term in the $r$-th product in Eq. (2.16). Having obtained Eq. (2.18), we can then write down the accurate formula of the boundary-sum operation $d_{n-1}$.

## 2.2 Extended group cohomology

We are now in the position to formulate the cohomology we found in Sec. 2.1 in a purely algebraic manner without referring to the geometric background. We shall allow general extended groups instead of merely finite ones. We shall also generalize the target from $U(1)$ to a general Abelian group $M$. We exclusively use the additive convention to describe the group composition of $M$. The identity of $M$ is denoted by 0.

### 2.2.1 Cochain complex

We begin with cochains, which correspond to lattice lagrangians.

**Definition 2.3.** For a positive integer $n$ and an Abelian group $M$, an *M-valued n-cochain* on an extended group $G$ is a map

$$
c_n : \prod_{k=1}^{n} G_k^{[n:k]} \mapsto M. \tag{2.19}
$$

The set of all $M$-valued $n$-cochains of $G$ will be denoted by $C^n(G, M)$ and is naturally an Abelian group induced by $M$.

Recall that sets $[n\!:\!k]$ are defined in Def. 2.2 and $Y^X$ is the standard notation for the set of all maps $X \mapsto Y$. Let us enumerate the looks of low-degree cochains: (to make things clear, we separate components by $|$ and $\|$ to avoid a messy hell of commas)

$$c_1\Big(\underbrace{g_1}_{G_1}\Big),\tag{2.20a}$$

$$c_2\Big(\underbrace{g_1 \,|\, g_2}_{G_1} \,\Big\|\, \underbrace{g_{1,2}}_{G_2}\Big),\tag{2.20b}$$

$$c_3\Big(\underbrace{g_1 \,|\, g_2 \,|\, g_3}_{G_1} \,\Big\|\, \underbrace{g_{1,2} \,|\, g_{1,3} \,|\, g_{2,3}}_{G_2} \,\Big\|\, \underbrace{g_{1,2,3}}_{G_3}\Big),\tag{2.20c}$$

$$c_4\Big(\underbrace{g_1 \,|\, g_2 \,|\, g_3 \,|\, g_4}_{G_1} \,\Big\|\, \underbrace{g_{1,2} \,|\, g_{1,3} \,|\, g_{2,3} \,|\, g_{1,4} \,|\, g_{2,4} \,|\, g_{3,4}}_{G_2} \,\Big\|\, \underbrace{g_{1,2,3} \,|\, g_{1,2,4} \,|\, g_{1,3,4} \,|\, g_{2,3,4}}_{G_3} \,\Big\|\, \underbrace{g_{1,2,3,4}}_{G_4}\Big).\tag{2.20d}$$

$g_{j_1,j_2,\cdots,j_k}$ gives an element of $G_k$ and recall that we exclusively stick to the increasing order $j_1 < j_2 < \cdots < j_k$. The total number of the components in $c_n$'s argument is superficially $2^n - 1$. However, since we put a dimension upper bound in Def. 1.1, it gradually increases polynomially with $n$.

We now formulate the differentials. Let us first formally introduce the pullback (2.18) under the new name of "face maps".

**Definition 2.4.** Given positive integers $n$ and $k \le n$, an integer $j$ with $0 \le j \le n$, and a group $H$, we define the *face maps*,

$$\partial_j : H^{[n:k]} \mapsto H^{[n-1:k]},\tag{2.21}$$

to be

$$(\partial_j h)_{q_1,q_2,\cdots,q_k} \equiv \begin{cases} h_{q_1,\cdots,q_\bullet,q_{\bullet+1}+1,\cdots,q_k+1}\,, & q_\bullet < j < q_{\bullet+1}\,, \\ h_{q_1,\cdots,q_\bullet,q_{\bullet+1}+1,\cdots,q_k+1} \, h_{q_1,\cdots,q_\bullet+1,q_{\bullet+1}+1,\cdots,q_k+1}\,, & q_\bullet = j\,, \end{cases}\tag{2.22}$$

where $1 \le q_1 < q_2 < \cdots < q_k \le n-1$.

We explicitly enumerate the face maps with $n \le 5$ in Table 1 of Appendix A. We need the following property of the face maps to ensure that our construction will be well-defined.

**Lemma 2.5.** *The face maps $\partial_j$ satisfy the identity*

$$\partial_i \partial_j = \partial_{j-1} \partial_i\,, \qquad if \ \ i < j\,.\tag{2.23}$$

*Proof.* We verify it directly from the definition (2.22). $\qquad\square$

We now define the differentials using the face maps and the Stokes-law sign (2.2).

**Definition 2.6.** Given a positive integer $n$, an Abelian group $M$, and an extended group $G$, the *$n$-th differential map*

$$\mathrm{d}_n : C^n(G, M) \mapsto C^{n+1}(G, M) \tag{2.24}$$

is the group homomorphism defined by

$$\mathrm{d}_n c_n \Big( g^{(1)} \Big\| g^{(2)} \Big\| \cdots \Big\| g^{(n)} \Big\| g^{(n+1)} \Big) \;\equiv\; \sum_{j=0}^{n+1} (-)^j \, c_n \Big( \partial_j g^{(1)} \Big\| \partial_j g^{(2)} \Big\| \cdots \Big\| \partial_j g^{(n)} \Big), \tag{2.25}$$

where $g^{(k)} \in G_k^{[n+1:k]}$.

We present the explicit expressions for these differentials with dimension $\leq 4$ in Appendix A, which can be quickly read off from Table 1. We now justify that the differentials we defined above are legitimate in the sense of homological algebra.

**Lemma 2.7.** *For an extended group $G$ and an Abelian group $M$, the cochains defined in Def. 2.3 and the differentials defined in Def. 2.6 assemble to a connective cochain complex,*

$$M \xrightarrow{\mathrm{d}_0} C^1(G, M) \xrightarrow{\mathrm{d}_1} C^2(G, M) \xrightarrow{\mathrm{d}_2} C^3(G, M) \xrightarrow{\mathrm{d}_3} C^4(G, M) \xrightarrow{\mathrm{d}_4} \cdots, \tag{2.26}$$

*where $\mathrm{d}_0$ is the trivial map. In other words, $\mathrm{d}_{n+1}\mathrm{d}_n = 0$ for any integer $n \geq 0$.*

*Proof.* The cases of $n = 0$ is trivial. For $n > 0$, the definition of differentials gives

$$\mathrm{d}_{n+1}\mathrm{d}_n c_n \Big( g^{(1)} \Big\| \cdots \Big\| g^{(n+2)} \Big) = \sum_{j=0}^{n+2} (-)^j \, \mathrm{d}_n c_n \Big( \partial_j g^{(1)} \Big\| \cdots \Big\| \partial_j g^{(n+1)} \Big),$$
$$= \sum_{i=0}^{n+1} \sum_{j=0}^{n+2} (-)^{i+j} \, c_n \Big( \partial_i \partial_j g^{(1)} \Big\| \cdots \Big\| \partial_i \partial_j g^{(n)} \Big) \tag{2.27}$$

There are in total $(n+2)(n+3)$ summands. We divide them into two parts, $i < j$ and $i \geq j$. Each part contains exactly $(n+2)(n+3)/2$ summands. Using Lemma 2.5, we can show that the $(i, j)$-term in the first part cancels the $(j-1, i)$-term in the second part, i.e.,

$$(-)^{i+j} \, c_n \Big( \partial_i \partial_j g^{(1)} \Big\| \cdots \Big\| \partial_i \partial_j g^{(n)} \Big) \;=\; - \, (-)^{(j-1)+i} \, c_n \Big( \partial_{j-1} \partial_i g^{(1)} \Big\| \cdots \Big\| \partial_{j-1} \partial_i g^{(n)} \Big). \tag{2.28}$$

Since each term in the first part cancels a distinct term in the second part, we can conclude that $\mathrm{d}_{n+1}\mathrm{d}_n = 0$ for any $n$. Namely, Eq. (2.26) does prescribe a cochain complex. $\qquad \square$

### 2.2.2 Cohomology and a Hurewicz-type corollary

Having obtained a cochain complex, we can now formulate its cohomology.

**Definition 2.8.** The cohomology of the cochain complex (2.26) is called the *cohomology of the extended group $G$ with coefficients in $M$*. More precisely, for a non-negative integer $n$, the $n$-th cohomology group of $G$ with coefficients in $M$ is defined as the quotient group

$$H^n(G, M) \;\equiv\; \frac{\ker \mathrm{d}_n}{\operatorname{im} \mathrm{d}_{n-1}}, \tag{2.29}$$

where $\operatorname{im} \mathrm{d}_{-1} \equiv \{0\}$. As usual, the elements of $\ker \mathrm{d}_n$ are called *$M$-valued $n$-cocycles* and the elements of $\operatorname{im} \mathrm{d}_{n-1}$ are called *$M$-valued $n$-coboundaries*.

It follows immediately that $H^0(G, M) = M$. When all the higher-form symmetries are set trivial, the extended group cohomology reduces to the original group cohomology of $G_1$; one may want to check the explicit expressions in Appendix A. In addition, there is another series of special cases that can be immediately evaluated.

**Corollary 2.9.** *If $G$ is an extended group such that $G_\bullet$ is trivial for all $\bullet < m$, then*

$$H^\bullet(G, M) = 0\,, \qquad \bullet < m; \tag{2.30a}$$

$$H^m(G, M) = \mathrm{Hom}_{\mathbb{Z}}(G_m, M)\,. \tag{2.30b}$$

*Proof.* In this case, all the $\bullet$-cochains with $\bullet < m$ are constant cochains, i.e.,

$$C^\bullet(G, M) \ \simeq \ M\,. \tag{2.31}$$

Because $\mathrm{d}_\bullet$ is defined via an alternating sum with $(\bullet+2)$ summands, $\mathrm{d}_\bullet$ is trivial for even $\bullet < m$, is an isomorphism for odd $\bullet < m-1$, and is a monomorphism for odd $\bullet = m-1$. Hence we always have $\ker \mathrm{d}_\bullet = \mathrm{im}\,\mathrm{d}_{\bullet-1}$ for all $\bullet < m$.

The probably nontrivial $m$-cochains only depend on $G_m$ and are functions on

$$G_m^{[m:m]} \ \simeq \ G_m\,. \tag{2.32}$$

Namely, they are univariate functions on $G_m$. Their $m$-cocycle condition has the argument

$$G_m^{[m+1:m]} \ \simeq \ G_m^{m+1}\,. \tag{2.33}$$

Thus we may relabel an $m$-tuple by its complementary 1-tuple with respect to $m+1$. After this relabelling, the $m$-cocycle condition looks like

$$c_m(g_1) - c_m(g_2 g_1) + c_m(g_3 g_2) - \cdots + (-)^m c_m(g_{m+1} g_m) + (-)^{m+1} c_m(g_{m+1}) = 0\,. \tag{2.34}$$

When $m$ is odd, Eq. (2.34) holds if and only if $c_m$ is a group homomorphism. Since now $\mathrm{d}_{m-1}$ is trivial, we obtain Eq. (2.30b). When $m$ is even, Eq. (2.34) holds if and only if $c_m$ is a group homomorphism up to a constant $m$-cochain. Since now $\mathrm{im}\,\mathrm{d}_{m-1}$ precisely consists of constant $m$-cochains, we still obtain Eq. (2.30b). $\qquad\square$

We hope this corollary reminds the readers of the cohomology version of the Hurewicz theorem: For an $(m-1)$-connected topological space $X$, its cohomology groups vanish below degree $m$, and at degree $m$ we have $H^m(X, M) = \mathrm{Hom}_{\mathbb{Z}}\big(\pi_m(X), M\big)$. Indeed, Corollary 2.9 is a scrawled omen of forthcoming Theorem 3.3.

## 2.3   Low-dimensional illustration

We now illustrate the lattice lagrangian of low-dimensional invertible field theories. The extended group $G$ will always be assumed finite.

### 2.3.1  $H^1\big(G, U(1)\big)$

A 1-cochain depends only on $G_1$ and is given by Eq. (2.20a):

$$c_1\Big(\underbrace{g_1}_{G_1}\Big)\,. \tag{2.35}$$

Accordingly, the valid content of $G$ in 1-dimensional spacetime is just the 0-form symmetry $G_1$. On a 1-simplex, according to Ansatz (2.8), the $G$ gauge field is specified by

$$0\bullet \xrightarrow{\;g_1\;} \bullet 1 \quad . \tag{2.36}$$

A $U(1)$-valued 1-cochain then gives the lagrangian of a 1-dimensional invertible field theory on this 1-simplex.

According to Def. 2.6, the 1-cocycle condition is

$$c_1\Big(g_2\Big) - c_1\Big(g_1 g_2\Big) + c_1\Big(g_1\Big) \;=\; 0\,. \tag{2.37}$$

It just says that $c_1$ is a group homomorphism and we simultaneously reach ordinary group cohomology of $G_1$ and Corollary 2.9. We can easily observe the geometric meaning of the 1-cocycle condition (2.37): The sum of a lagrangian over the boundary of a 2-simplex is trivial if we arrange the gauge fields as follows, i.e.,

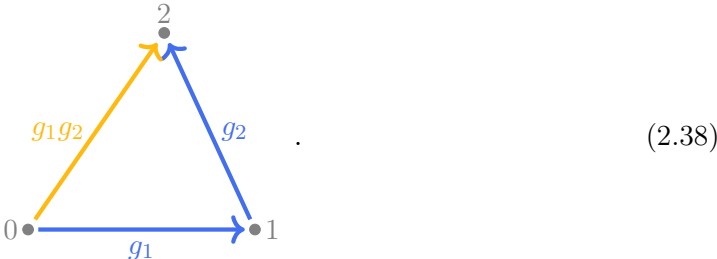

$$\tag{2.38}$$

Each of the three 1-facets of this 2-simplex corresponds to a distinct term in the 1-cocycle condition (2.37). This indeed gives a locally flat gauge field on the entire 2-simplex.

### 2.3.2  $H^2\big(G, U(1)\big)$

Things start to get interesting from this dimension. A 2-cochain is given by Eq. (2.20b):

$$c_2\Big(\underbrace{g_1\,\big|\,g_2}_{G_1}\,\Big\|\,\underbrace{g_{1,2}}_{G_2}\Big)\,. \tag{2.39}$$

Accordingly, the valid content of $G$ in 2-dimensional spacetime only contains 0-form symmetry $G_1$ and 1-form symmetry $G_2$. We now assign the $G$ gauge fields to a 2-simplex

according to Ansatz (2.8). The result is

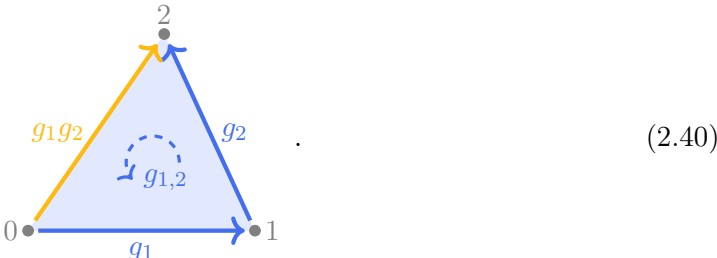

$$(2.40)$$

The agreement between the 2-simplex gauge field (2.40) and the geometric 1-cocycle condition (2.38) checks our derivation in Sec. 2.1.2.

A $U(1)$-valued 2-cochain then gives the lagrangian of a 2-dimensional invertible field theory on this 2-simplex. According to Def. 2.6, the 2-cocycle condition is

$$c_2\Big(g_2 \,\big|\, g_3 \,\big\|\, g_{2,3}\Big) - c_2\Big(g_1 g_2 \,\big|\, g_3 \,\big\|\, g_{1,3} g_{2,3}\Big)$$
$$+ \, c_2\Big(g_1 \,\big|\, g_2 g_3 \,\big\|\, g_{1,2} g_{1,3}\Big) - c_2\Big(g_1 \,\big|\, g_2 \,\big\|\, g_{1,2}\Big) \;=\; 0\,. \tag{2.41}$$

When $G_2$ is trivial, this reduces to the 2-cocycle condition for $G_1$'s group cohomology. When $G_1$ is trivial, we again reach Corollary 2.9. The geometric meaning of the 2-cocycle condition (2.41) is also clear: The sum of lagrangian over the boundary of a 3-simplex is trivial if we arrange the gauge fields in the following way, i.e.,

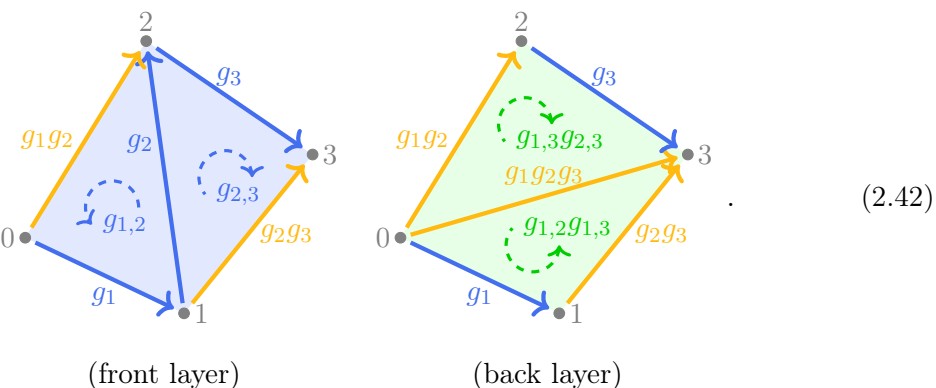

$$(2.42)$$

(front layer)         (back layer)

Each of the four 2-facets of this 3-simplex corresponds to a distinct term in the 2-cocycle condition (2.41). This is indeed a locally flat gauge field on the entire 3-simplex.

### 2.3.3   $H^3\big(G, U(1)\big)$

Things start to get complicated from this dimension. A 3-cochain is given by Eq. (2.20c):

$$c_3\Big(\underbrace{g_1 \,\big|\, g_2 \,\big|\, g_3}_{G_1} \,\Big\|\, \underbrace{g_{1,2} \,\big|\, g_{1,3} \,\big|\, g_{2,3}}_{G_2} \,\Big\|\, \underbrace{g_{1,2,3}}_{G_3}\Big)\,. \tag{2.43}$$

Accordingly, the valid content of $G$ in 3-dimensional spacetime only contains 0-form symmetry $G_1$, 1-form symmetry $G_2$, and 2-form symmetry $G_3$. We now assign the $G$ gauge

fields to a 3-simplex according to Ansatz (2.8). The result is

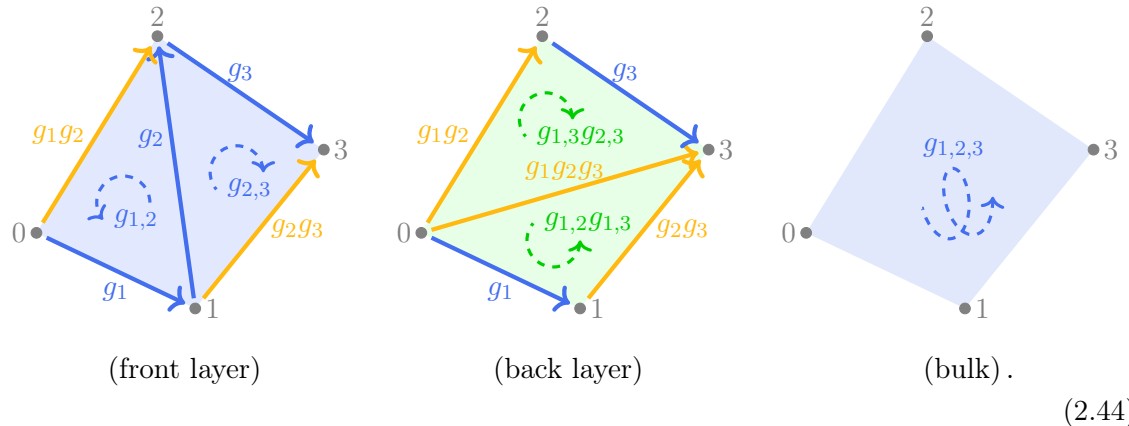

$$(2.44)$$

Again, the agreement between the 3-simplex gauge field (2.44) and the geometric 2-cocycle condition (2.42) checks our derivation in Sec. 2.1.2.

A $U(1)$-valued 3-cochain then gives the lagrangian of a 3-dimensional invertible field theory on this 3-simplex. According to Def. 2.6, the 3-cocycle condition is

$$
\begin{aligned}
& c_3\Big( g_2 \,\big|\, g_3 \,\big|\, g_4 \,\big\|\, g_{2,3} \,\big|\, g_{2,4} \,\big|\, g_{3,4} \,\big\|\, g_{2,3,4} \Big) \\
& - c_3\Big( g_1 g_2 \,\big|\, g_3 \,\big|\, g_4 \,\big\|\, g_{1,3} g_{2,3} \,\big|\, g_{1,4} g_{2,4} \,\big|\, g_{3,4} \,\big\|\, g_{1,3,4} g_{2,3,4} \Big) \\
& + c_3\Big( g_1 \,\big|\, g_2 g_3 \,\big|\, g_4 \,\big\|\, g_{1,2} g_{1,3} \,\big|\, g_{1,4} \,\big|\, g_{2,4} g_{3,4} \,\big\|\, g_{1,2,4} g_{1,3,4} \Big) \\
& - c_3\Big( g_1 \,\big|\, g_2 \,\big|\, g_3 g_4 \,\big\|\, g_{1,2} \,\big|\, g_{1,3} g_{1,4} \,\big|\, g_{2,3} g_{2,4} \,\big\|\, g_{1,2,3} g_{1,2,4} \Big) \\
& + c_3\Big( g_1 \,\big|\, g_2 \,\big|\, g_3 \,\big\|\, g_{1,2} \,\big|\, g_{1,3} \,\big|\, g_{2,3} \,\big\|\, g_{1,2,3} \Big) \quad = \quad 0 \,.
\end{aligned}
$$

$$(2.45)$$

When $G_2, G_3$ are trivial, this reduces to the cocycle condition for $G_1$'s group cohomology. When $G_1, G_2$ are trivial, we again recover Corollary 2.9. The geometric meaning of the 3-cocycle condition (2.45) is that the sum of lagrangian over the boundary of a 4-simplex is trivial if the gauge field is arranged according to Ansatz (2.8), which ensures locally-flatness on the 4-simplex.

## 2.4 Remark

A few remarks follow.

### Choice of ansatz

Ansatz (2.8) is not the unique choice. In principle, other ansätze are also allowed, such as the one we used to count the degrees of freedom in Sec. 2.1.1. One may even abandon any ansatz and just work with the constraint (2.5), as what Eilenberg and MacLane did in early years [61–63]. Then the cochains and the differentials will take a very different form, but the resulting cohomology would still be isomorphic.

Our Ansatz (2.8), nevertheless, has quite a few advantages over other ansätze or no ansatz. Our cochains and differentials take pretty tractable forms and generalize the familiar

group cohomology in a natural and intuitive way. In particular, the face maps (2.22) involve multiplications of at most two group elements only. As we will see in Sec. 3, our ansatz also has a direct connection to a particular construction of Eilenberg-MacLane spaces and their products, the Dold-Kan correspondence, which is standard in simplicial homotopy theory.

**Generalized Dijkgraaf-Witten-Yetter model**

We can obtain a topological field theory by dynamically gauging the global symmetry of an invertible field theory with finite symmetry. The partition function $e^{i\mathcal{S}}$ of the invertible field theory becomes the weight in a path integral over all gauge fields. Physically, this means obtaining a topological order by dynamically gauging the global symmetry of a symmetry protected topological phase. We call these topological field theories the generalized Dijkgraaf-Witten-Yetter models. They are examples of fully-extended topological field theories (see e.g. [64]).

From the angle of dynamical gauging, Dijkgraaf and Witten [59] gauged 0-form symmetry and Yetter [65] gauged 2-group symmetry. The lattice formulation is also useful to study the generalized Dijkgraaf-Witten-Yetter models. It benefits the evaluations of partition functions, Hilbert spaces, and higher-categorical data. It also benefits the evaluation of the spectrum of topological operators and their anomalies. People have considered and described the lattice formulation for 0-form symmetries [59], 1-form symmetries [66], and 2-group symmetries [8]. Our findings can be used to construct the lattice formulation for the generalized Dijkgraaf-Witten-Yetter models with extended-group gauge symmetries.

## 3   Equivalence of algebraic and topological approaches

For invertible field theories, the algebraic approach characterizes and classifies lagrangians $\mathcal{L}$ by extended group cohomology, $H^\bullet\big(G, U(1)\big)$ (see Def. 2.8). The topological approach characterizes and classifies actions $\mathcal{S}$ by cohomology of the classifying space, $H^\bullet\big(\mathcal{B}G, U(1)\big)$ (see Def. 3.2 for $\mathcal{B}G$). There is a rather straightforward connection (2.11) between them, namely $\mathcal{S} = \sum \mathcal{L}$. We thus expect an equivalence between the algebraic approach and the topological approach.

In this section, we shall present a rigorous proof of $H^\bullet(G, M) = H^\bullet(\mathcal{B}G, M)$. Following Ref. [59], we can readily construct an algebraic cochain on $G$ from a singular cochain on $\mathcal{B}G$. Nevertheless, it is highly nontrivial to show that such a construction eventually induces an isomorphism between the two different cohomologies. An essentially equivalent result was first stated by Eilenberg and MacLane [61]; they provided a proof for the case of group cohomology only, though. The proof we shall exhibit is based on a now standard theoretical toolkit, simplicial homotopy theory.

If we assumed the readers' familiarity with simplicial homotopy theory, we could conclude this section almost immediately. However, to convince more readers, we shall merely assume the familiarity with the basics of general and algebraic topology, and also make our exposition as self-contained as possible. Thus this section can also be regarded as an introduction to simplicial homotopy theory. Our main references are Refs. [60, 67] for algebraic topology and Refs. [68–71] for simplicial homotopy.

### 3.1   Isomorphism theorem

The classifying space of an individual discrete $(m-1)$-form symmetry $H$ is the Eilenberg-MacLane space $K(H, m)$, also usually denoted by the delooping notation $B^m H$ (especially in physics literature), which is uniquely characterized by its homotopy groups [60, 67].

**Definition 3.1.** Consider a group $H$ and a positive integer $m$, and assume $H$ is Abelian when $m > 1$. A space $W$ is called an *Eilenberg-MacLane space* $K(H, m)$ if (i) $W$ has the homotopy type of a CW complex; (ii) $W$ is path-connected; (iii) for any positive integer $\bullet$,

$$\pi_\bullet(W, *) = \begin{cases} H, & \bullet = m \\ \{1\}, & \bullet \neq m \end{cases}, \tag{3.1}$$

for a basepoint $* \in W$.

CW complexes [72], invented by Whitehead, are the well-behaved class of spaces under the consideration of algebraic topology. We direct the readers to [60, Proposition 4.30] for a proof of the uniqueness of $K(H, m)$ up to homotopy equivalence. The classifying space of an extended group is the product of all the individual classifying spaces.

**Definition 3.2.** The *classifying space* $\mathcal{B}G$ of an extended group $G$ is given by

$$\mathcal{B}G \equiv \prod_{k=1}^{\infty} K(G_k, k) \tag{3.2}$$

up to homotopy equivalence.

This is actually a finite product in disguise since we put a dimension cutoff in Def. 1.1; this is exactly the reason why we did that. Our goal in this section is to prove the following isomorphism theorem.

**Theorem 3.3.** *For an extended group $G$, its cohomology (Def. 2.8) is isomorphic to the cohomology of its classifying space. Namely, for any non-negative integer $n$ and any Abelian group $M$, there is a group isomorphism*

$$H^n(G, M) = H^n(\mathcal{B}G, M). \tag{3.3}$$

*Proof.* Combine Theorem 3.13 (plus Def. 3.14), Theorem 3.22, and Theorem 3.35. □

This whole section is devoted to presenting this proof, which includes three steps.

**(I)** We construct a series of topological spaces $|\mathcal{K}(H, m)|$ by gluing numerous simplices in appropriate ways. This corresponds to Sec. 3.2 and Theorem 3.13 (plus Def. 3.14).

**(II)** We evaluate the cohomology of $\prod_k |\mathcal{K}(G_k, k)|$ to show its isomorphism to extended group cohomology. This corresponds to Sec. 3.3 and Theorem 3.22.

**(III)** We show that the spaces we have constructed are Eilenberg-MacLane spaces, i.e., $|\mathcal{K}(H, m)|$ is a $K(H, m)$. This corresponds to Sec. 3.4 and Theorem 3.35.

Our construction for $|\mathcal{K}(H, m)|$ will not be new in essence because it actually matches the simplicial Abelian group yielded by the Dold-Kan correspondence (assuming Abelian $H$ even if $m = 1$). The Dold-Kan theorem is a result in simplicial homotopy theory and we shall say a bit more about it in Sec. 3.5.

## 3.2 Step I: Simplicial construction

Despite the early appearance, a simplicial complex or a $\Delta$-complex behaves terribly for many purposes of algebraic topology. Given this, Eilenberg and Zilberg invented simplicial sets aimed at providing an ideal purely combinatory approach to homotopy [73].

### 3.2.1 Simplicial set

We want to construct a topological space by gluing numerous simplices. Let us first introduce the building blocks and the elementary gluing manipulations.

**Definition 3.4.** For a non-negative integer $n$, the *standard $n$-simplex* $\Delta^n$ is defined by

$$\Delta^n \equiv \left\{ (t_0, t_1, \cdots, t_n) \,\middle|\, \sum_{i=0}^n t_i = 1 \text{ and } t_i \geq 0 \text{ for all } i \right\} \subseteq \mathbb{R}^{n+1}. \tag{3.4}$$

For any integer $i$ with $0 \leq i \leq n$, the *inclusion maps* $\delta^i : \Delta^{n-1} \mapsto \Delta^n$ and the *collapse maps* $\sigma^i : \Delta^{n+1} \mapsto \Delta^n$ are respectively defined by

$$\delta^i(t_0, t_1, \cdots, t_{n-1}) \equiv (t_0, \cdots, t_{i-1}, 0, t_i, \cdots, t_{n-1}), \tag{3.5a}$$

$$\sigma^i(t_0, t_1, \cdots, t_{n+1}) \equiv (t_0, \cdots, t_{i-1}, t_i + t_{i+1}, t_{i+2} \cdots, t_{n+1}). \tag{3.5b}$$

The inclusion map $\delta^i$ attaches $\Delta^{n-1}$ onto $\Delta^n$ as the facet opposite to the $i$-vertex. The collapse map $\sigma^i$ shrinks $\Delta^{n+1}$ into $\Delta^n$ along the $(i, i+1)$-edge. They are the elementary gluing manipulations we shall use. They satisfy the following property.

**Lemma 3.5.** *The inclusion and the collapse maps satisfy the* cosimplicial identities,

$$\delta^j \delta^i = \delta^i \delta^{j-1}, \qquad i < j, \tag{3.6a}$$

$$\sigma^j \sigma^i = \sigma^i \sigma^{j+1}, \qquad i \leq j, \tag{3.6b}$$

$$\sigma^j \delta^i = \begin{cases} \delta^i \sigma^{j-1}, & i < j \\ \mathrm{id}, & i = j \ or \ j+1 \\ \delta^{i-1} \sigma^j, & i > j+1 \end{cases} . \tag{3.6c}$$

*Proof.* We verify them just by direct evaluations. $\qquad\qquad\square$

Given building blocks and elementary manipulations, we also need a blueprint to tell us how to organize them. Different blueprints will lead us to different topological spaces. Such a blueprint is called a simplicial set [68–71].

**Definition 3.6.** A *simplicial set* $X$ contains a sequence of sets $X_0, X_1, X_2, \cdots$ together with the *face maps* $\delta_i : X_n \to X_{n-1}$ and the *degeneracy maps* $\sigma_i : X_n \to X_{n+1}$ for all $i = 0, 1, \cdots, n$, such that the following *simplicial identities* are satisfied:

$$\delta_i \delta_j = \delta_{j-1} \delta_i, \qquad i < j, \tag{3.7a}$$

$$\sigma_i \sigma_j = \sigma_{j+1} \sigma_i, \qquad i \leq j, \tag{3.7b}$$

$$\delta_i \sigma_j = \begin{cases} \sigma_{j-1} \delta_i, & i < j \\ \mathrm{id}, & i = j \text{ or } j+1 \\ \sigma_j \delta_{i-1}, & i > j+1 \end{cases} . \tag{3.7c}$$

These simplicial identities are the dual of the cosimplicial identities (3.6).

In this blueprint, $X_n$ specifies how many $n$-simplices we use. The maps indicate how to organize the elementary manipulations to glue building blocks. The simplicial identities are designed to accommodate the cosimplicial identities of elementary gluing manipulations to make the resulting space have perfectly ideal behaviors.

### 3.2.2 Geometric realization

Let us accurately describe how to construct a topological space from a simplicial set [68–71].

**Definition 3.7.** For a simplicial set $X$, its *geometric realization* $|X|$ is a space defined by

$$|X| \equiv \coprod_{n=0}^{\infty} X_n \times \Delta^n \Big/ \sim \tag{3.8}$$

where $\sim$ is the equivalence relation generated by

$$(x, \delta^i p) \sim (\delta_i x, p) \quad \text{for} \quad x \in X_n, p \in \Delta^{n-1}, \tag{3.9a}$$

$$(\sigma_i x, p) \sim (x, \sigma^i p) \quad \text{for} \quad p \in \Delta^n,\, x \in X_{n-1}. \tag{3.9b}$$

Here $X_n$ is endowed with the discrete topology and $\coprod$ denotes the disjoint union.

Although the disjoint union seems big, the equivalence classes are usually also big and the resulting space has a pretty reasonable size. To gain more insights into $|X|$, we consider an alternative description of the above equivalence relation.

**Lemma 3.8.** *Let $\rhd$ be the transitive relation on $\coprod_{n=0}^{\infty} X_n \times \Delta^n$ generated by*

$$(y, q) \rhd (\delta_i y, \sigma^i q) \quad \text{if} \quad y \in \operatorname{im} \sigma_i \text{ or } q \in \operatorname{im} \delta^i. \tag{3.10}$$

*Then $\rhd$ generates the equivalence relation $\sim$ in Def. 3.7. Furthermore, $a \sim b$ if and only if $a = b$ or $a \rhd b$ or $a \lhd b$ or $a \rhd c \lhd b$ for some $c$.*

*Proof.* Using $\sigma^i \delta^i = \mathrm{id}$ and $\delta_i \sigma_i = \mathrm{id}$ from the third (co)simplicial identity, we see that the relations (3.9) and (3.10) are equivalent. Hence $\rhd$ generates $\sim$. We now come to the next part. Extensively using all (co)simplicial identities, we can show that $(\delta_i y, \sigma^i q) \lhd (y, q) \rhd (\delta_j y, \sigma^j q)$ implies (we assume $i \geq j$ without loss of generality)

$$\begin{cases} (\delta_i y, \sigma^i q) = (\delta_j y, \sigma^j q), & i = j \\ (\delta_i y, \sigma^i q) = (\delta_j y, \sigma^j q), & i = j + 1 \text{ with } y \in \operatorname{im} \sigma_j, q \in \operatorname{im} \delta^i \\ (\delta_i y, \sigma^i q) \rhd (\delta_j \delta_i y, \sigma^j \sigma^i q) \lhd (\delta_j y, \sigma^j q), & i = j + 1 \text{ with all other possibilities} \\ (\delta_i y, \sigma^i q) \rhd (\delta_j \delta_i y, \sigma^j \sigma^i q) \lhd (\delta_j y, \sigma^j q), & i > j + 1 \end{cases} \tag{3.11}$$

Let us consider an equivalence between $a, b \in \coprod_{n=0}^{\infty} X_n \times \Delta^n$, which is a finite chain made by elementary $\lhd$ and $\rhd$ in some order. Once we see a fragment $\cdots f \lhd g \rhd h \cdots$ in this chain, using Eq. (3.11), we can replace it by $\cdots f \cdots$ or $\cdots f \rhd g' \lhd h \cdots$. Iterating this procedure, we will eventually obtain a chain of elementary relations in the form

$$a \rhd f_1 \rhd \cdots \rhd f_k \lhd \cdots \lhd f_N \lhd b \tag{3.12}$$

where both the numbers of $\rhd$'s and $\lhd$'s might be zero. Depending on zero or nonzero, we reach $a = b$ or $a \rhd b$ or $a \lhd b$ or $a \rhd f_k \lhd b$. $\qquad\square$

This characterization of the equivalence relation is illuminating: Every equivalence class has a unique lowest-dimensional point and we can "$\rhd$" all other points to it. This lowest-dimensional point is the perfect representative for this equivalence class.

**Definition 3.9.** If $X$ is a simplicial set, then $x \in X_n$ is called *non-degenerate* if it is not the image of any $\sigma_j$. Moreover, $(x, p) \in X_n \times \Delta^n$ is called *non-degenerate* if $x$ is non-degenerate and $p$ lies in the interior (i.e., not the image of any $\delta^i$).

**Theorem 3.10.** *$X$ is a simplicial set. In $\coprod_{n=0}^{\infty} X_n \times \Delta^n$, non-degenerate points bijectively represent the equivalence classes generated by Eq. (3.9).*

*Proof.* A non-degenerate point cannot appear on the left of $\triangleright$ defined in Lemma 3.8. Then Lemma 3.8 tells us that $a \sim b$ implies $a = b$ when $a$ and $b$ are both non-degenerate. We thus proved injectivity. As for surjectivity, let us consider $a \in X_n \times \Delta^n$. If $a$ degenerates, we can always find a non-degenerate $b \in X_{n-m} \times \Delta^{n-m}$ for a sufficiently large $m$ such that $a \triangleright b$ simply due to the dimension lower bound $n - m \geq 0$. We hence proved surjectivity. $\qquad\square$

Given this theorem, one may want to construct geometric realization solely from non-degenerate elements. Then such a construction can hardly be combinatory.

**Corollary 3.11.** *If $X$ is a simplicial set, then $|X|$ is a CW complex with one $n$-cell for each non-degenerate element of $X_n$.*

*Proof.* The cell structure immediately follows Theorem 3.10, the closure finiteness (C) follows the finite number of $\delta_i$'s at each dimension, and the weak topology (W) follows the general-topology property of quotient maps. $\qquad\square$

### 3.2.3 Construction of $\mathcal{K}(H, m)$

We now construct the simplicial set $X$ whose geometric realization is going to be $K(H, m)$. The correct recipe is not hard to guess. First, $X_n$ ought to be the trivial group $\{1\}$ when $n < m$ and $H^{[n:m]}$ when $n \geq m$, respectively (symbol $[n{:}m]$ is defined in Def. 2.2). Second, the face maps ought to be those defined in Def. 2.4 as we already call them "face maps" there. Finally, we need to find the degeneracy maps and show the simplicial identities.

**Definition 3.12.** Given positive integers $n$ and $k \leq n$, an integer $j$ with $0 \leq j \leq n$, and a group $H$, we define the map

$$s_j : H^{[n:k]} \mapsto H^{[n+1:k]} \tag{3.13}$$

to be

$$(s_j h)_{q_1, q_2, \cdots, q_k} \equiv \begin{cases} h_{q_1, \cdots, q_\bullet, q_{\bullet+1}-1, \cdots, q_k-1}, & q_\bullet < j+1 < q_{\bullet+1}, \\ 1, & j+1 \in \{q_\bullet\}, \end{cases} \tag{3.14}$$

where $1 \leq q_1 < q_2 < \cdots < q_k \leq n+1$ and $1$ denotes the identity element of $H$.

**Theorem 3.13.** *Consider a group $H$ and a positive integer $m$, and assume $H$ is Abelian if $m > 1$. Then the following data,*

$$X_n \equiv \begin{cases} \{1\}, & 0 \leq n < m \\ H^{[n:m]}, & n \geq m \end{cases}, \tag{3.15a}$$

$$\delta_j : X_n \mapsto X_{n-1} \equiv \begin{cases} \mathbf{1}, & 1 \leq n \leq m \\ \partial_j, & n > m \end{cases}, \quad \sigma_j : X_n \mapsto X_{n+1} \equiv \begin{cases} \mathbf{1}, & 0 \leq n < m \\ s_j, & n \geq m \end{cases}, \tag{3.15b}$$

*assemble to a simplicial set. Here $\partial_j$ is defined in Def. 2.4, $s_j$ is defined in Def. 3.12, and $\mathbf{1}$ denotes the trivial map that sends everything to the identity element.*

*Proof.* We verify the simplicial identities just by directly evaluating them. $\qquad\square$

**Definition 3.14.** We use $\mathcal{K}(H, m)$ to denote the simplicial set defined by Theorem 3.13.

Though perhaps not immediately obvious, this is identical to what can be deduced from the Dold-Kan correspondence (see Sec. 3.5). Given Corollary 3.11, let us inspect the non-degenerate elements of $\mathcal{K}(H, m)$ to see the cell structure of $|\mathcal{K}(H, m)|$. First, we have a unique element $* \in \mathcal{K}(H, m)_0$, contributing a 0-cell. $\mathcal{K}(H, m)_n$ with $0 < n < m$ contains $\sigma_0^n *$ only, i.e., there is no $n$-cell with $0 < n < m$. Every element in $\mathcal{K}(H, m)_m$ except $\sigma_0^m *$ is non-degenerate, i.e., there are as many $m$-cells as $H - \{1\}$. When $H$ is not trivial, higher dimensions are harder to track but we can readily see a bunch of non-degenerate elements, i.e., there are a bunch of $n$-cells with $n > m$. This cell structure echoes a methodology of constructing Eilenberg-MacLane spaces called "attaching higher-dimensional cells to kill higher homotopy groups" (see [60, Sec. 4.2]). We conclude this discussion with a cute result.

**Lemma 3.15.** *If $H$ is the trivial group, then $|\mathcal{K}(H, m)|$ is a point for any $m$.*

*Proof.* In this case, every $\mathcal{K}(H, m)_n$ contains just one element $\sigma_0^n *$. Then Theorem 3.10 tells us that $|\mathcal{K}(H, m)|$ is just a point. □

### 3.3 Step II: Cohomology evaluation

We want to evaluate the cohomology of $\prod_k |\mathcal{K}(G_k, k)|$. The strategy is to convert everything into computations on simplicial sets. Namely, we want to convert product and cohomology into structures on simplicial sets. Let us start with cohomology. It is actually pretty easy to construct a cochain complex from each simplicial set.

**Lemma 3.16.** *$X$ is a simplicial set and $M$ is an Abelian group. The following data,*

$$M^{X_0} \xrightarrow{\mathrm{d}_0} M^{X_1} \xrightarrow{\mathrm{d}_1} M^{X_2} \xrightarrow{\mathrm{d}_2} M^{X_3} \xrightarrow{\mathrm{d}_3} M^{X_4} \xrightarrow{\mathrm{d}_4} \cdots , \tag{3.16}$$

*where*

$$\mathrm{d}_n \equiv \sum_{j=0}^n (-)^j \, \delta_j^* , \qquad \delta_j^*(X_n \mapsto M) \equiv X_{n+1} \overset{\delta_j}{\mapsto} X_n \mapsto M , \tag{3.17}$$

*assemble to a cochain complex.*

*Proof.* The proof is completely identical to that of Lemma 2.7 and makes use of the first simplicial identity only. □

**Definition 3.17.** The cohomology of the cochain complex defined by Lemma 3.16 is denoted by $H^\bullet(X, M)$.

If a space is the geometric realization $|X|$ of some simplicial set $X$, there are two distinct cohomologies associated to it. The first is the topological $H^\bullet(|X|, M)$, usually defined as the singular cohomology (or directly via the Eilenberg–Steenrod axioms), which depends only on the homotopy type of $|X|$ only. The second is the simplicial $H^\bullet(X, M)$, which a priori depends on $X$. Fortunately, the two cohomologies can be proven isomorphic.

**Theorem 3.18.** *For a simplicial set $X$ and any Abelian group $M$, there are group isomorphisms for all non-negatives $n$,*

$$H^n(X, M) = H^n(|X|, M) . \tag{3.18}$$

*Proof.* This is one of the key results in simplicial homotopy theory[3] and its proof is not as elementary as others in this paper. Therefore, instead of presenting a proof here, we direct the readers to the well-established literature [68, Sec. 16][4]. Alternatively, one may prove this by establishing the relative $H^\bullet(X, Y; M)$ and verify the Eilenberg–Steenrod axioms. $\square$

Having learned how to dealing with cohomology, we now turn to products.

**Definition 3.19.** $X$ and $Y$ are simplicial sets. Their *product* $X \times Y$ is a simplicial set defined by the following data:

$$(X \times Y)_n \equiv X_n \times Y_n\,, \qquad \delta_i(x, y) \equiv (\delta_i x, \delta_i y)\,, \qquad \sigma_i(x, y) \equiv (\sigma_i x, \sigma_i y)\,. \tag{3.19}$$

The simplicial identities are satisfied in an obvious way.

Milnor [74] first discovered that, thanks to degeneracy maps, this disturbingly simple-minded definition works perfectly. To show this, we need a pretty technical geometric property of collapse maps that cannot be deduced solely from cosimplicial identities.

**Lemma 3.20.** *If $p \in \Delta^\alpha$ and $q \in \Delta^\beta$ are interior points, then there are two unique ordered strings of non-negative integers $i_1 < \cdots < i_K$ and $j_1 < \cdots < j_M$ such that $\{i_\bullet\} \cap \{j_\bullet\} = \varnothing$ and $p = \sigma^{i_1} \cdots \sigma^{i_K} r$ and $q = \sigma^{j_1} \cdots \sigma^{i_M} r$ for some $r$.*

*Proof.* We prove it by finding the solution. Suppose $p = (p_0, p_1, \cdots, p_\alpha) \in \Delta^\alpha \subseteq \mathbb{R}^{\alpha+1}$ and $q = (q_0, q_1, \cdots, q_\beta) \in \Delta^\beta \subseteq \mathbb{R}^{\beta+1}$. For every $a = 0, 1, \cdots, \alpha$ and $b = 0, 1, \cdots, \beta$, we define

$$u_a \equiv \sum_{\ell=0}^{a} p_\ell\,, \qquad v_b \equiv \sum_{\ell=0}^{b} q_\ell\,. \tag{3.20}$$

Let $0 < t_0 < \cdots < t_N = 1$ be the ordered strings of all elements in $\{u_\bullet\} \cup \{v_\bullet\}$. Clearly, $N \leq \alpha + \beta$. Then let $i_1 < \cdots < i_K$ be those $i$'s such that $t_i \notin \{u_\bullet\}$ and let $j_1 < \cdots < j_M$ be those $j$'s such that $t_j \notin \{v_\bullet\}$. Let $r \equiv (t_0, t_1 - t_0, t_2 - t_1, \cdots, t_N - t_{N-1}) \in \Delta^N \subseteq \mathbb{R}^{N+1}$. We can then verify $p = \sigma^{i_1} \cdots \sigma^{i_K} r$ and $q = \sigma^{j_1} \cdots \sigma^{i_M} r$. The uniqueness comes from the theory of linear equations. $\square$

**Theorem 3.21.** *$X$ and $Y$ are simplicial sets. Then there is a homeomorphism*

$$|X \times Y| \cong |X| \times |Y| \tag{3.21}$$

*if every $X_n$ and every $Y_n$ are at most countable[5].*

---

[3]Do not confuse this with another theorem in algebraic topology. The simplicial cohomology defined on simplicial complexes or $\Delta$-complexes takes an identical form to Theorem 3.16. But the difference is crucial: Simplicial complexes or $\Delta$-complexes do not make use of degeneracy maps.

[4]We note that [68, Sec. 16] does not directly contain a proof of the cohomology version. It does give a proof of the homology version [68, Proposition 16.2]. One may either modify that proof to adapt cohomology or derive the cohomology version based on it.

[5]The countability assumption can be dropped but the price is that we must assign $|X| \times |Y|$ a topology finer than the product topology. Systematically, we should do product in the category **CGHaus** (compactly generated Hausdorff spaces) rather than **Top** (spaces). The former is Cartesian closed while the latter is not. For our purpose, we do not need such a generality.

*Proof.* Because the canonical projection $\mathrm{pr}_n$ commutes with every $\delta_i$ and $\sigma_i$, $\mathrm{pr}_n \times \mathrm{id}_{\Delta^n} :$ $(X \times Y)_n \times \Delta^n \mapsto X_n \times \Delta^n$ induces a continuous surjective map $|X \times Y| \mapsto |X|$ . Similarly we have $|X \times Y| \mapsto |Y|$. Altogether we obtain a continuous surjective map $\eta : |X \times Y| \mapsto$ $|X| \times |Y|$. To prove injectivity, we trace the non-degenerate representatives under $\eta$ (see Theorem 3.10). Let us consider a non-degenerate $(x, y, r) \in X_n \times Y_n \times \Delta^n$. Given the second simplicial identity, let us assume $x = \sigma_{i_K} \cdots \sigma_{i_1} \bar{x}$ with $i_1 < \cdots < i_K$ for non-degenerate $\bar{x}$ and $y = \sigma_{j_M} \cdots \sigma_{j_1} \bar{y}$ with $j_1 < \cdots < j_M$ for non-degenerate $\bar{y}$. Then $\{i_\bullet\} \cap \{j_\bullet\} = \varnothing$ because once $i_k = j_m$, $(x, y)$ cannot be non-degenerate. The non-degenerate representative of $[(x, r)]$ is $(\bar{x}, p)$ with $p = \sigma^{i_1} \cdots \sigma^{i_K} r$ and the non-degenerate representative of $[(y, r)]$ is $(\bar{y}, q)$ with $q = \sigma^{j_1} \cdots \sigma^{i_M} r$. Lemma 3.20 says that $p$ and $q$ uniquely determine $r$, $i_\bullet$'s, and $j_\bullet$'s. We thus obtain $\eta^{-1}$ and proved injectivity.

We need the countability assumption to show the continuity of $\eta^{-1}$. Since $\eta^{-1}$ is clearly continuous on each product cell of $|X| \times |Y|$, it is continuous if $|X| \times |Y|$ has the weak topology, i.e., if $|X| \times |Y|$ is a CW complex. Hence this is in fact a general-topology problem about CW complexes. As proved in [60, Theorem A.6], the product of two CW complexes is also a CW complex if there are at most countably many cells. $\square$

After all, we now have all the tools to evaluate the cohomology of $\prod_k |\mathcal{K}(G_k, k)|$.

**Theorem 3.22.** *$G$ is an extended group and $M$ is an Abelian group. Then there is a group isomorphism for each non-negative integer $n$,*

$$H^n(G, M) = H^n \left( \prod_{k=1}^{\infty} \left| \mathcal{K}(G_k, k) \right|, M \right) . \tag{3.22}$$

*Proof.* Combining the dimension cutoff $N$ in Def. 1.1 and Lemma 3.15, we see that the infinite product above is actually a finite product, i.e.,

$$\text{r.h.s.} = H^n \left( \prod_{k=1}^{N} \left| \mathcal{K}(G_k, k) \right|, M \right) . \tag{3.23}$$

Due to Def. 1.1 and Def. 3.14, $\mathcal{K}(G_k, k)_\ell$ is at most countable for all $\ell$ and $k$. Hence Theorem 3.21 is applicable. Iterating Theorem 3.21 $N$ times and then applying Theorem 3.18, we transform the cohomology of a space into the cohomology of a simplicial set, i.e.,

$$\text{r.h.s.} = H^n \left( \prod_{k=1}^{N} \mathcal{K}(G_k, k), M \right) . \tag{3.24}$$

We can readily see that $\left( \prod_{k=1}^{N} \mathcal{K}(G_k, k) \right)_\ell$ is exactly the argument for the $\ell$-cochains defined in Def. 2.3. Also, we can readily see that the face maps of $\prod_{k=1}^{N} \mathcal{K}(G_k, k)$ are precisely those used in Def. 2.6. Therefore, when we evaluate the cochain complex of $\prod_{k=1}^{N} \mathcal{K}(G_k, k)$ defined by Theorem 3.16, we simply recover the same cochain complex given by Lemma 2.7. We thus proved the theorem. $\square$

### 3.4 Step III: Space identification

We finally come to the last step to show $|\mathcal{K}(H,m)|$ is indeed a $K(H,m)$. Since Theorem 3.11 ensures that $|\mathcal{K}(H,m)|$ is a CW complex, what we need to do is to compute the homotopy groups of $|\mathcal{K}(H,m)|$. Just like cohomology and product, we also want to evaluate $|X|$'s homotopy groups by computing something about $X$.

#### 3.4.1 Kan condition and homotopy group

This time, however, we cannot achieve the above goal for arbitrary simplicial sets. We can only do this for simplicial sets that satisfy the Kan condition [68–71].

**Definition 3.23.** $X$ is a simplicial set. For any non-negative integers $n$ and $k \leq n+1$, an $(n,k)$-*horn on $X$* is a map

$$\lambda : \ \{0, \cdots, k{-}1, k{+}1, \cdots, n{+}1\} \ \mapsto \ X_n \tag{3.25}$$

such that $\delta_i \lambda_j = \delta_{j-1} \lambda_i$ for all $i < j$ and $i, j \neq k$.

**Definition 3.24.** If $\lambda$ is an $(n,k)$-horn on $X$, then $y \in X_{n+1}$ is called a *filler of $\lambda$* if $\lambda_i = \delta_i y$ for all $i \neq k$. A simplicial set $X$ is called *fibrant* (or satisfies the *Kan condition*) if every horn on $X$ has a filler.

The geometric interpretation of the Kan condition is clear: If all but one $(n{-}1)$-facets of an $n$-simplex are there, the entire $n$-simplex including the rest $(n{-}1)$-facet is also there. We shall soon show that $\mathcal{K}(H,m)$ is fibrant. We now introduce based homotopy on simplicial sets [68–71].

**Definition 3.25.** $X$ is a simplicial set and $* \in X_0$. For $a, b \in X_n$, some $y \in X_{n+1}$ is called a *homotopy from $a$ to $b$ based at $*$* if for all $i = 0, 1, \cdots, n{+}1$,

$$\delta_i y = \begin{cases} \sigma_0^n *, & i < n \\ a, & i = n \\ b, & i = n+1 \end{cases} . \tag{3.26}$$

If such $y$ exists, we say $a$ and $b$ are *homotopic based at $*$*. In particular, when $a, b \in X_0$, the base $*$ does not actually appear in the definition and we thus drop "based at $*$".

The geometric intuition is that an $n$-simplex provides a based homotopy between two adjacent $(n{-}1)$-facets by shrinking all other $(n{-}1)$-facets to the basepoint. If we further shrink the rest two $(n{-}1)$-facets to the basepoint, we are left with a based sphere $S^n$. We are thus led to the following definition for the set of based $n$-spheres.

**Definition 3.26.** $X$ is a simplicial set and $* \in X_0$. For a positive integer $n$, we define

$$Z_n(X, *) \equiv \left\{ x \in X_n \, \middle| \, \delta_i x = \sigma_0^{n-1} * \text{ for all } i = 0, 1, \cdots, n \right\}. \tag{3.27}$$

As the condition above is vacuously true when $n = 0$, we also define $Z_0(X) = X_0$.

**Lemma 3.27.** $Z_n(X, *) \neq \varnothing$ *since it at least contains* $\sigma_0^n *$.

*Proof.* We verify $\delta_i \sigma_0^n * = \sigma_0^{n-1} *$ by repeating the third simplicial identity. $\qquad \square$

Armed with the Kan condition, we can find based homotopy classes of spheres.

**Lemma 3.28.** *For a fibrant simplicial set* $X$, *being homotopic is an equivalence relation on* $Z_0(X)$. *Moreover, being homotopic based at* $*$ *is an equivalence relation on* $Z_n(X, *)$ *for any positive integer* $n$ *and any* $* \in X_0$.

*Proof.* $Z_0(X)$ and $Z_n(X, *)$ are treated in the same footing so we just write $Z_n$ and allow $n$ to be any non-negative integer. We verify reflexivity by observing that for any $a \in Z_n$, $\sigma_n a$ is a homotopy from $a$ to itself. Recall that given reflexivity, both symmetry and transitivity are implied by $a \sim b$, $a \sim c \Rightarrow b \sim c$. To prove this, consider $a, b, c \in Z_n$, a homotopy $y$ from $a$ to $b$, and a homotopy $y'$ from $a$ to $c$. Then the data

$$\lambda_0 \equiv \lambda_1 \equiv \cdots \equiv \lambda_{n-1} \equiv \sigma_0^{n+1} *, \qquad \lambda_n \equiv y, \qquad \lambda_{n+1} \equiv y' \tag{3.28}$$

define a $(n+1, n+2)$-horn $\lambda$ on $X$. If $\lambda$ has a filler $z \in X_{n+2}$, then $\delta_{n+2} z$ is a homotopy from $b$ to $c$. $\qquad \square$

**Definition 3.29.** For a fibrant simplicial set $X$ and $* \in X_0$, we define the sets

$$\pi_0(X) \equiv Z_0(X)/\sim \tag{3.29a}$$
$$\pi_n(X, *) \equiv Z_n(X, *)/\sim, \qquad n > 0 \tag{3.29b}$$

where $\sim$ is the equivalence relation given by Lemma 3.28.

We now convert these sets into groups in a way akin to the proof of Lemma 3.28.

**Definition 3.30.** $X$ is a fibrant simplicial set and $* \in X_0$. For $a, b \in Z_n(X, *)$, a *compositor* of $(a, b)$ is a filler of the $(n, n)$-horn $\lambda$ on $X$ defined by

$$\lambda_0 \equiv \lambda_1 \equiv \cdots \equiv \lambda_{n-2} \equiv \sigma_0^n *, \qquad \lambda_{n-1} \equiv a, \qquad \lambda_{n+1} \equiv b. \tag{3.30}$$

We usually write an unspecified compositor of $(a, b)$ as either $\mathcal{C}_{a,b}$ or $\mathcal{C}(a, b)$.

**Theorem 3.31.** *The map* $[a] \times [b] \mapsto [\delta_n \mathcal{C}_{a,b}]$ *makes* $\pi_n(X, *)$ *a group (with identity* $[\sigma_0^n *]$).

*Proof.* We first show that this map is well-defined. We can verify $\delta_n \mathcal{C}_{a,b} \in Z_n(X, *)$ using the first simplicial identity. Then let us show the independence of the choice of a compositor. Let us consider another compositor $\mathcal{C}'_{a,b}$. Then the data

$$\Lambda_0 \equiv \cdots \equiv \Lambda_{n-2} \equiv \sigma_0^{n+1} *, \qquad \Lambda_{n-1} \equiv \sigma_n a, \qquad \Lambda_{n+1} \equiv \mathcal{C}_{a,b}, \qquad \Lambda_{n+2} \equiv \mathcal{C}'_{a,b} \tag{3.31}$$

define an $(n+1, n)$-horn $\Lambda$. If $\Lambda$ has a filler $x \in X_{n+2}$, then $\delta_n x$ is a homotopy from $\delta_n \mathcal{C}_{a,b}$ to $\delta_n \mathcal{C}'_{a,b}$. Next, let us show the independence of the choice of $b \in [b]$. Let us consider another

$b' \in [b]$, along with a homotopy $w \in X_{n+1}$ from $b'$ to $b$, and a compositor $\mathcal{C}_{a,b'}$. Then the data

$$\Omega_0 \equiv \cdots \equiv \Omega_{n-2} \equiv \sigma_0^{n+1}*\,, \qquad \Omega_{n-1} \equiv \sigma_{n-1}x\,, \qquad \Omega_n \equiv \mathcal{C}_{a,b'}\,, \qquad \Omega_{n+2} \equiv w \qquad (3.32)$$

define an $(n{+}1,n{+}1)$-horn $\Omega$. If $\Omega$ has a filler $y \in X_{n+2}$, then $\delta_{n+1}y$ is a compositor of $(a,b)$. Because of the compositor-independence we have just proved, we see $\delta_n \mathcal{C}_{a,b'} = \delta_n \delta_{n+1} y \sim \delta_n \mathcal{C}_{a,b}$. This proves the $b$-independence and we can prove the $a$-independence in a similar way.

   We now check the group axioms and start from associativity. Let us consider $a, b, c \in Z_n(X, *)$ and compositors $\mathcal{C}_{a,b}$, $\mathcal{C}_{b,c}$, and $\mathcal{C}(\delta_n \mathcal{C}_{a,b}, c)$. Then the data

$$\Gamma_0 \equiv \cdots \equiv \Gamma_{n-2} \equiv \sigma_0^{n+1}*\,, \quad \Gamma_{n-1} \equiv \mathcal{C}_{a,b}\,, \quad \Gamma_{n+1} \equiv \mathcal{C}(\delta_n \mathcal{C}_{a,b}, c)\,, \quad \Gamma_{n+2} \equiv \mathcal{C}_{b,c} \quad (3.33)$$

define an $(n{+}1,n)$-horn $\Gamma$. If $\Gamma$ has a filler $z \in X_{n+2}$, then $\delta_n z$ is a $\mathcal{C}(a, \delta_n \mathcal{C}_{b,c})$. Hence $\delta_n \delta_n z = \delta_n \mathcal{C}(\delta_n \mathcal{C}_{a,b}, c)$ proves associativity. We now come to the existence of identity and inverse. Let us consider any $a \in Z_n(X, *)$. Then any compositor $\mathcal{C}_{\sigma_0^n *, a}$ is a homotopy from $\delta_n \mathcal{C}_{\sigma_0^n *, a}$ to $a$. This makes $[\sigma_0^n *]$ a left identity. Besides, the data

$$\xi_0 \equiv \cdots \equiv \xi_{n-2} \equiv \xi_n \equiv \sigma_0^n *\,, \qquad \xi_{n+1} \equiv a \qquad (3.34)$$

define an $(n, n{-}1)$-horn $\xi$. If $\xi$ has a filler $w \in X_{n+1}$, then $w$ is a $\mathcal{C}(\delta_{n-1}w, a)$. Hence $\delta_n w = \sigma_0^n *$ means that $[\delta_{n-1}w]$ is a left inverse of $[a]$. Recall that associativity, left identity, and left inverse imply the whole group axioms. $\qquad \square$

   After heavy combinatory endeavors, we finally obtain something like homotopy groups on simplicial sets. We would be disappointed if they are not isomorphic to the real homotopy groups of the geometric realization.

**Theorem 3.32.** *For a fibrant simplicial set $X$, we have a bijection,*

$$\pi_0(X) = \pi_0(|X|) \qquad (3.35)$$

*and group isomorphisms*

$$\pi_n(X, *) = \pi_n(|X|, |*|) \qquad (3.36)$$

*for any positive integer $n$ and any $* \in X_0$.*

*Proof.* This is one of the key results in simplicial homotopy theory and its proof is not as elementary as others in this paper. Hence instead of presenting a proof, we direct the readers to the well-established literature [68, Sec. 16], especially [68, Theorem 16.5]. $\qquad \square$

### 3.4.2   Identifying $|\mathcal{K}(H,m)|$ as Eilenberg-MacLane

To apply what we have just learned, we first check the Kan condition on $\mathcal{K}(H,m)$.

**Theorem 3.33.** $\mathcal{K}(H,m)$ *is fibrant.*

*Proof.* We shall explicitly find a filler for every $(n,k)$-horn $\lambda$ on $\mathcal{K}(H,m)$. First, $\lambda$ trivially has a filler when $n < m$. We then turn to $n \geq m$ and construct a filler $x \in \mathcal{K}(H,m)_{n+1} = H^{[n+1:m]}$. Let us define the components of $x$ by two finite inductions. The first induction begins with $\{q_\bullet\} \in [n+1:m]$ such that $1 \notin \{q_\bullet\}$, for which we define

$$x_{q_1,\cdots,q_m} \equiv (\lambda_0)_{q_1-1,\cdots,q_m-1}. \tag{3.37}$$

We then define $x_{q_1,\cdots,q_m}$ such that $2 \notin \{q_\bullet\}$ but $1 \in \{q_\bullet\}$ through

$$x_{q_1,\cdots,q_m} \equiv (\lambda_1)_{q_1,q_2-1,\cdots,q_m-1}\, x^{-1}_{q_1+1,q_2,\cdots,q_m}, \qquad q_1 = 1. \tag{3.38}$$

Given that we have defined $x_{q_1,\cdots,q_m}$ such that $j \notin \{q_\bullet\}$ but all $1,2,\cdots,j-1 \in \{q_\bullet\}$, we next define $x_{q_1,\cdots,q_m}$ such that $j+1 \notin \{q_\bullet\}$ but $1,2,\cdots,j \in \{q_\bullet\}$ through

$$x_{q_1,\cdots,q_m} \equiv (\lambda_j)_{q_1,\cdots,q_\bullet,q_{\bullet+1}-1,\cdots,q_m-1}\, x^{-1}_{q_1+1,\cdots,q_\bullet+1,q_{\bullet+1},\cdots,q_m}, \qquad q_\bullet = j \tag{3.39}$$

This induction stops at $j = k-1$. The second induction begins with $\{q_\bullet\} \in [n+1:m]$ such that $n+1 \notin \{q_\bullet\}$ excluding those already in the first induction, for which we define

$$x_{q_1,\cdots,q_m} \equiv (\lambda_{n+1})_{q_1,\cdots,q_m}. \tag{3.40}$$

We then define $x_{q_1,\cdots,q_m}$ such that $n \notin \{q_\bullet\}$ but $n+1 \in \{q_\bullet\}$, excluding those already defined in the first induction, through

$$x_{q_1,\cdots,q_m} \equiv x^{-1}_{q_1,\cdots,q_{m-1},q_m-1}\, (\lambda_n)_{q_1,\cdots,q_{m-1},q_m-1}, \qquad q_m = n+1. \tag{3.41}$$

Given that we have defined $x_{q_1,\cdots,q_m}$ such that $j \notin \{q_\bullet\}$ but $j+1,j+2,\cdots,n+1 \in \{q_\bullet\}$, we next define $x_{q_1,\cdots,q_m}$ such that $j-1 \notin \{q_\bullet\}$ but $j,j+1,\cdots,n+1 \in \{q_\bullet\}$, excluding those already defined in the first induction, through

$$x_{q_1,\cdots,q_m} \equiv x^{-1}_{q_1,\cdots,q_\bullet,q_{\bullet+1}-1,\cdots,q_m-1}\, (\lambda_{j-1})_{q_1,\cdots,q_\bullet,q_{\bullet+1}-1,\cdots,q_m-1}, \qquad q_{\bullet+1} = j. \tag{3.42}$$

This induction stops at $j = k+2$. The two finite inductions exhaust all indices in $[n+1:m]$ because a remained index would contain all $1,2,\cdots,n+1$, which cannot happen due to $n+1 > m$. We can then verify that $x$ is indeed a filler of $\lambda$ via direct evaluation based on Eq. (2.22). $\qquad\square$

Let us now compute the homotopy groups.

**Theorem 3.34.** *We have a bijection*

$$\pi_0\big(\mathcal{K}(H,m)\big) = \{1\}, \tag{3.43}$$

*and group isomorphisms for all $n = 1,2,\cdots,$*

$$\pi_n\big(\mathcal{K}(H,m),*\big) = \begin{cases} H, & n = m \\ \{1\}, & n \neq m \end{cases} \tag{3.44}$$

*with $* \in \mathcal{K}(H,m)_0$.*

*Proof.* Eq. 3.43 is obvious since $\mathcal{K}(H, m)_0$ has only one element $*$. To find the homotopy groups, let us evaluate $Z_n\big(\mathcal{K}(H, m), *\big)$ which we abbreviate as $Z_n$. When $n < m$, $Z_n$ is trivial since $\mathcal{K}(H, m)_n$ also has only one element. $Z_m = \mathcal{K}(H, m)_m = H^{[m:m]}$ as sets because $\mathcal{K}(H, m)_{m-1}$ is trivial. To show $Z_n$ is trivial when $n > m$, we first notice that $\sigma_0^n *$ is the identity element of $H^{[n:m]}$. Then let us consider $h \in H^{[n:m]}$. For any $\{q_1, \cdots, q_m\} \in [n:m]$, due to $n > m$, we can always find an integer $j \neq q_1, \cdots, q_m$ but $1 \leq j \leq n$. According to Eq. (2.22), $\delta_j h = \sigma_0^{n-1} *$ implies $h_{q_1, \cdots, q_m} = 1$. Therefore, $h \in Z_n$ implies that all of $h$'s components are 1. That is to say, $Z_n$ contains only the trivial $\sigma_0^n *$ when $n > m$. We hence proved the triviality of $\pi_n\big(\mathcal{K}(H, m), *\big)$ for all $n \neq m$.

Now we focus on the only nontrivial $Z_m = H^{[m:m]}$. Since $H^{[m:m]} \sim H$, we shall drop the index of its elements. Also, for $x \in \mathcal{K}(H, m)_{m+1} = H^{[m+1:m]} \sim H^{m+1}$, we shall relabel its index, which is an $m$-tuple, by its complementary 1-tuple with respect to $m+1$. We did the same thing in the proof of Corollary 2.9. Now, if $x \in \mathcal{K}(H, m)_{m+1}$ is a homotopy between $a, b \in Z_m$, then Eq. (2.22) implies

$$x_1 = x_2 x_1 = \cdots = x_m x_{m-1} = 1, \qquad x_{m+1} x_m = a, \qquad x_{m+1} = b, \qquad (3.45)$$

which has a solution if and only if $a = b$. Hence $a \sim b$ implies $a = b$, i.e., each element in $Z_m$ belongs to a distinct homotopy class. We finally check the group structure. We can verify that $y \in \mathcal{K}(H, m)_{m+1}$ defined by the components

$$y_1 = \cdots = y_{m-1} = 1, \qquad y_m = a, \qquad y_{m+1} = b. \qquad (3.46)$$

is a compositor of $(a, b)$. Because $\delta_m y = ba$, we obtain a group composition $[a] \times [b] \mapsto [ba]$ according to Theorem 3.31. Hence $\pi_m\big(\mathcal{K}(H, m), *\big) = H^{op}$, the opposite group of $H$. Recall that $H^{op}$ is isomorphic to $H$ via $h \mapsto h^{-1}$, or via $h \mapsto h$ when $H$ is Abelian. $\qquad \square$

**Theorem 3.35.** $|\mathcal{K}(H, m)|$ *is an Eilenberg-MacLane space* $K(H, m)$.

*Proof.* Corollary 3.11 says $|\mathcal{K}(H, m)|$ is a CW-complex. Combining Theorems 3.32, 3.33, and 3.34, we see that $|\mathcal{K}(H, m)|$ is path-connected and its homotopy groups do satisfy the requirement of Def. 3.1. Therefore, we conclude that $|\mathcal{K}(H, m)|$ is a $K(H, m)$. $\qquad \square$

### 3.5 Remark

A few remarks follow.

### Simplicial Abelian group and Dold-Kan correspondence

In a simplicial set, if all the sets are groups and all the maps are group homomorphisms, we obtain the notion of a simplicial group. Similarly, we can have a simplicial Abelian group, a simplicial topological space, and so on. To some extent, the structure of a simplicial Abelian group is completely understood through the Dold-Kan correspondence [70, Corollary 2.3], which says that there is an equivalence between the category of simplicial Abelian groups and the category of connective chain complexes. The homotopy groups (Theorem 3.32) of a simplicial Abelian group are isomorphic to the homology groups of its corresponding

chain complex via the Dold-Kan correspondence [70, Corollary 2.7]. Moreover, the geometric realization of any simplicial Abelian group is always a product of Eilenberg-MacLane spaces [70, Proposition 2.20].

$\mathcal{K}(H, m)$ we built is exactly a simplicial Abelian group except $m = 1$ with non-Abelian $H$. Its Dold-Kan correspondence is the chain complex such that only its $m$-th group is nontrivial and is $H$. More generally, for an extended group $G$ with Abelian $G_1$, the product $\prod_k \mathcal{K}(G_k, k)$ is also a simplicial Abelian group. Its Dold-Kan correspondence is the chain complex such that its $k$-th group is $G_k$ and all the differentials are trivial. $\mathcal{K}(H, 1)$ with non-Abelian $H$ becomes an exception not only because of its failure to be a simplicial Abelian group. Actually, it fails to be a simplicial group in the first place; one can check that the face maps defined in Eq. (2.22) are not group homomorphisms when $H$ is non-Abelian.

### $\infty$-groupoid and homotopy hypothesis

Our narrative centers on topological spaces and describes simplicial sets (Def. 3.6) merely as tools to construct spaces. Nevertheless, simplicial sets themselves make great sense. In particular, fibrant simplicial sets (Def. 3.24), also called *Kan complexes*, provide the most well-studied definition for $\infty$-groupoids. An $\infty$-groupoid ought to be an $\infty$-category whose $n$-morphisms are invertible up to equivalence for all $n$. More concretely, in a fibrant simplicial set $X$, $X_0$ specifies the objects, $X_n$ specifies the $n$-morphisms, and a composition of $n$-morphisms is provided by a horn-filler in a way that generalizes Def. 3.30 and Theorem 3.31. The geometric realization (Def. 3.7) eventually establishes a bijective correspondence between equivalence classes of $\infty$-groupoids and weak homotopy types of topological spaces. This is the homotopy hypothesis, which is a proved theorem when one defines $\infty$-groupoids by fibrant simplicial sets.

Given the triumph of fibrant simplicial sets, people found that, as long as we properly weaken the Kan condition (Def. 3.24), we can allow $n$-morphisms to be non-invertible while keeping all other desired features. In such a way, simplicial sets can provide a pretty good foundation for general higher-categories. This topological-stream approach to higher-category is usually dubbed quasi-category (see e.g. [75]).

# 4 't Hooft anomaly: Algebraic approach

In the case of finite extended-group symmetries and ordinary anomalies, anomaly inflow can be characterized in a very intuitive and precise manner via the algebraic approach. In this section, the extended group $G$ will always be assumed finite.

## 4.1 Anomaly inflow

Let us take a compact $(n+1)$-dimensional spacetime $\mathcal{M}$ and consider an invertible field theory on $\mathcal{M}$. In the discrete formulation, as we analyzed in Sec. 2.1.2, when $\mathcal{M}$ is closed, the $(n+1)$-cocycle condition ensures the triangulation-independence, i.e., gauge invariance. But this is longer true if $\partial \mathcal{M} \neq \varnothing$. Let us consider an elementary triangulation rearrangement that annihilates an $(n+1)$-simplex on the boundary $\partial \mathcal{M}$, as illustrated by

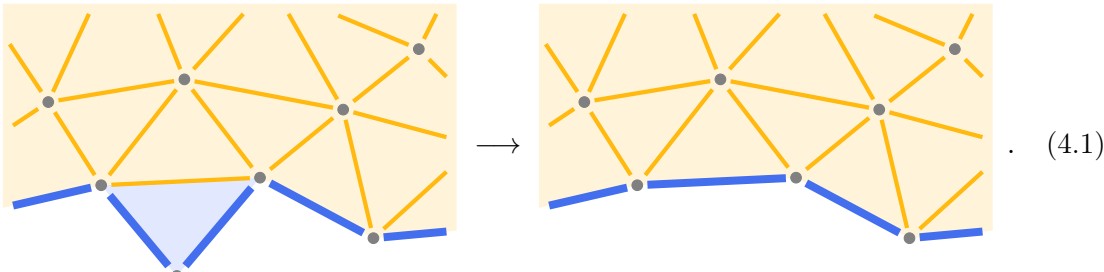

$$\hspace{6cm} . \hspace{1cm} (4.1)$$

As an $(n+1)$-simplex is annihilated, the action changes by one piece of lagrangian, i.e.,

$$\mathcal{S} \;\rightarrow\; \mathcal{S} - \mathcal{L} \,. \tag{4.2}$$

Hence gauge invariance is lost. To remedy it, we need to see that the $(n+1)$-simplex annihilation also causes a triangulation rearrangement on the boundary $\partial \mathcal{M}$. More precisely, let us suppose this $(n+1)$-simplex initially has $\alpha$ $n$-facets on the boundary. After annihilation, its other $\beta$ $n$-facets are instead exposed on the boundary, with $\alpha + \beta = n + 2$. If on the boundary lives an anomalous theory whose partition function changes as

$$\mathcal{Z} \rightarrow \mathcal{Z} \, \mathrm{e}^{\mathrm{i}\mathcal{L}} \,, \tag{4.3}$$

under the boundary triangulation rearrangement $\alpha \rightarrow \beta$, then gauge invariance is restored. The entire system, comprised of the $(n+1)$-dimensional invertible field theory on $\mathcal{M}$ and the $n$-dimensional anomalous theory on $\partial \mathcal{M}$, has an invariant partition function under the triangulation rearrangement, i.e.,

$$\mathcal{Z} \, \mathrm{e}^{\mathrm{i}\mathcal{S}} \;\rightarrow\; \mathcal{Z} \, \mathrm{e}^{\mathrm{i}\mathcal{S}} \,. \tag{4.4}$$

The anomalous theory itself need not be formulated discretely; we just describe its background gauge fields using the discrete formulation.

What we have described above is precisely anomaly inflow, which establishes a bijective correspondence between $n$-dimensional anomalies and $(n+1)$-dimensional invertible field theories. In this way, 't Hooft anomalies are characterized and classified by extended group cohomology. A few important remarks follow.

**(I)** We emphasize the compatibility among different types of elementary rearrangements. If we count $\alpha \to \beta$ and its inverse $\beta \to \alpha$ only once, there are $\lfloor \frac{n+2}{2} \rfloor$ types of elementary rearrangements. Even without referring to anomaly inflow, from the anomalous phase of one elementary rearrangement, we should still be able to derive that of another.

**(II)** In general, we can consider an interface between two invertible field theories. An elementary rearrangement causes $\mathcal{S}_1 \to \mathcal{S}_1 - \mathcal{L}_1$ and $\mathcal{S}_2 \to \mathcal{S}_2 + \mathcal{L}_2$ in the two regions and $\mathcal{Z} \to \mathcal{Z}\, e^{i(\mathcal{L}_1 - \mathcal{L}_2)}$ on the interface. This does not bring anything new due to invertibility. However, if we replace invertible field theories by symmetry protected topological phases, interfaces are more appropriate settings (see our discussion in Sec. 1).

**(III)** Many readers may be more familiar with the dual picture of what we have described. Let us dualize a triangulation to obtain a "cotriangulation" where a $k$-simplex is replaced by a codimension-$k$ polytope. $G_k$ elements now live on codimension-$k$ polytopes, which are nothing but the topological operators of the $(k-1)$-form symmetry $G_k$, i.e.,

$$k\text{-simplex with gauge field} \quad \longleftrightarrow \quad \text{codimension-}k \text{ topological operator} \qquad (4.5)$$

Hence our discrete formulation is exactly the same as a web of multidimensional topological operators (see Sec. 1). Anomalies are the extra phase factors accompanying rearrangements of the web, i.e., gauge transformations.

## 4.2 Low-dimensional illustration

We now use "cotriangulations" [Remark (III) above] to illustrate the general behavior of $n$-dimensional anomalies with $n \le 3$. Our discussion will be schematic rather than concrete. In what follows, $-^{\text{ab}}$ denotes the Abelianization.

### 4.2.1 1-dimensional anomaly

Let us start with 1-dimensional anomalous theories and 2-dimensional invertible field theories. Since $3 = 2 + 1$, there is only one type of elementary rearrangement $2 \rightleftharpoons 1$. Dualizing the 2-simplex (2.40) and cut out a 1-dimensional edge, we obtain the anomaly inflow,

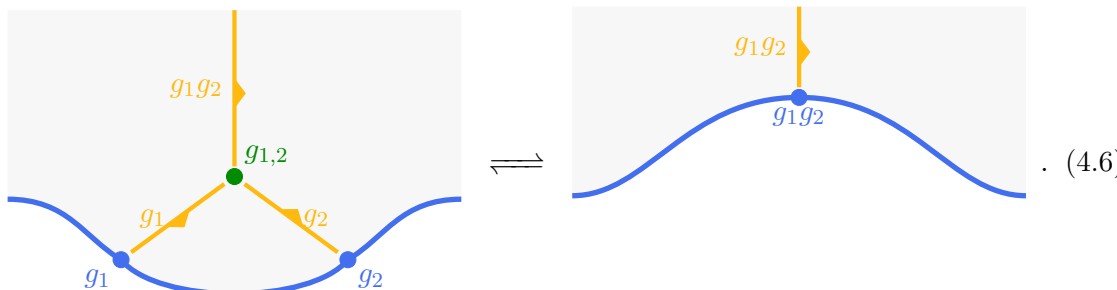

$$. \quad (4.6)$$

The 2nd cohomology of an extended group $G$ canonically splits in the following way,

$$H^2\Big(G, U(1)\Big) = H^2\Big(BG_1, U(1)\Big) \oplus_{\mathbb{Z}} \text{Hom}_{\mathbb{Z}}\Big(G_2, U(1)\Big). \qquad (4.7)$$

$H^2\big(BG_1, M\big)$ classifies central extensions of $G_1$ by $M$. Hence 0-form anomalies are well-known to mean a projective representation: We obtain an anomalous phase factor when we fuse $g_1$ and $g_2$ into $g_1g_2$ on the boundary.

1-form anomalies $\mathrm{Hom}_{\mathbb{Z}}\big(G_2, U(1)\big)$ are rather peculiar since 1-form symmetry is absent in 1-dimensional spacetime. Yet the anomaly inflow does exist and awaits to be received by a boundary theory. In general, extended group cohomology canonically splits as follows at any dimension,

$$H^{n+1}\big(G, U(1)\big) = \mathrm{Hom}_{\mathbb{Z}}\big(G_{n+1}, U(1)\big) \oplus_{\mathbb{Z}} \Big\{ \text{terms irrelevant to } G_{n+1} \Big\}. \tag{4.8}$$

Since no $n$-form symmetry is present in $n$-dimensional spacetime, the nature of boundary theories awaits to be clarified. We refer to this peculiar circumstance as anomaly overflow.

### 4.2.2 2-dimensional anomaly

We now turn to 2-dimensional anomalous theories and 3-dimensional invertible field theories. This dimension, having been extensively studied, is illuminating and yet simple. We have a canonical splitting,

$$H^3\big(G, U(1)\big) =$$
$$H^3\big(BG_1, U(1)\big) \oplus_{\mathbb{Z}} \mathrm{Hom}_{\mathbb{Z}}\big(G_1^{\mathrm{ab}} \otimes_{\mathbb{Z}} G_2, U(1)\big) \oplus_{\mathbb{Z}} \mathrm{Hom}_{\mathbb{Z}}\big(G_3, U(1)\big), \tag{4.9}$$

corresponding to 0-form anomalies, mixed 0&1-form anomalies, and overflow 2-form anomalies, accordingly.

Because $4 = 2 + 2 = 3 + 1$, there are two types of elementary rearrangements, namely $2 \rightleftharpoons 2$ and $3 \rightleftharpoons 1$. We give up drawing an illustration of anomaly inflow like Eq. (4.6). Instead, we directly draw the rearrangements of topological operators on the boundary. Decode Ansatz (2.8) into geometry, we obtain for $2 \rightleftharpoons 2$,

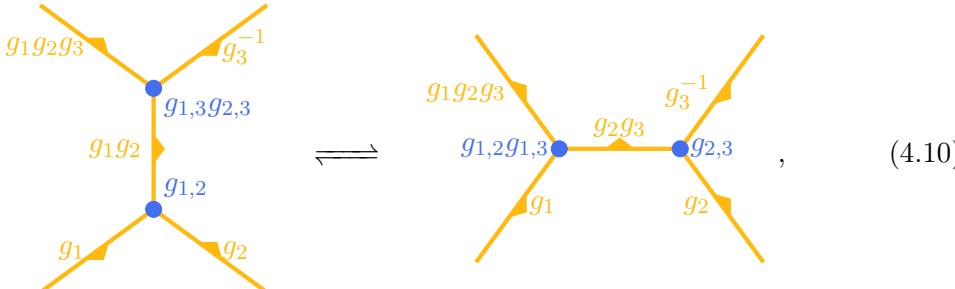

$$\tag{4.10}$$

and for $3 \rightleftharpoons 1$,

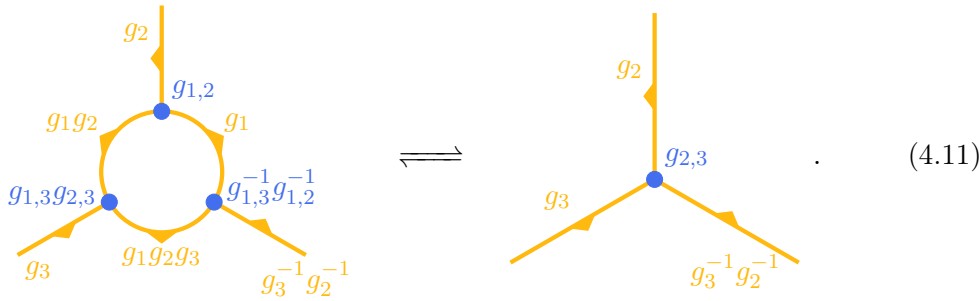

$$\tag{4.11}$$

0-form anomalies $H^3\big(BG_1, U(1)\big)$ are well-known. From the angle of non-invertible symmetry, a finite 0-form symmetry is described by a fusion category, i.e., a coherent set of F-symbols (see e.g. [76, 77]). Then the $2 \rightleftharpoons 2$ diagram (4.10) corresponds to the monoidal structure (i.e., associator) while the $3 \rightleftharpoons 1$ diagram (4.11) is related to the rigidity structure (i.e., quantum dimension). The axioms of fusion categories ensure them to give identical phase factors when the fusion rule is invertible.

The mixed 0&1-form anomalies $\mathrm{Hom}_{\mathbb{Z}}\big(G_1^{\mathrm{ab}} \otimes_{\mathbb{Z}} G_2,\, U(1)\big)$ are simpler and yet also interesting. The diagrams where merely $g_1$ and $g_{2,3}$ are nontrivial can detect these anomalies. Under this setting, the $2 \rightleftharpoons 2$ diagram says that we obtain an anomalous phase if we fuse a line topological operator with a point topological operator. The $3 \rightleftharpoons 1$ diagram says that we obtain an anomalous phase if we contract a line topological operator equipped with a point topological operator. These anomalies mean that the line and the point topological operators act mutually-projectively on the Hilbert spaces. The infrared of gapped phases matching these anomalies are 2-dimensional BF models.

In general, for any non-negative integers $p, q$ such that

$$p \neq q, \quad p + q = n - 1, \tag{4.12}$$

$H^{n+1}\big(G, U(1)\big)$ always contains a direct summand $\mathrm{Hom}_{\mathbb{Z}}\big(G_{p+1}^{\mathrm{ab}} \otimes_{\mathbb{Z}} G_{q+1}^{\mathrm{ab}},\, U(1)\big)$. This corresponds to mixed $p$&$q$-form anomalies in dimension $n$, implying that the codimension-$(p+1)$ and the codimension-$(q+1)$ topological operators act mutually-projectively on the Hilbert spaces. The infrared of gapped phases matching these anomalies are $n$-dimensional BF models with dynamical $p$-form and $q$-form gauge fields.

### 4.2.3   3-dimensional anomaly

We then come to 2-dimensional anomalous theories and 3-dimensional invertible field theories. This dimension starts to get intricate. We have a canonical splitting

$$H^4\Big(G, U(1)\Big) = \mathrm{Hom}_{\mathbb{Z}}\Big(G_1^{\mathrm{ab}} \otimes_{\mathbb{Z}} G_3,\, U(1)\Big) \oplus_{\mathbb{Z}} \mathrm{Hom}_{\mathbb{Z}}\Big(G_4,\, U(1)\Big)$$
$$\oplus_{\mathbb{Z}} H^4\Big(BG_1, U(1)\Big) \oplus_{\mathbb{Z}} H^4\Big(B^2 G_2, U(1)\Big) \oplus_{\mathbb{Z}} \mathrm{Hom}_{\mathbb{Z}}\Big(H_2\big(BG_1, G_2\big),\, U(1)\Big) \tag{4.13}$$

corresponding to mixed 0&2-form anomalies, overflow 3-form anomalies, 0-form anomalies, 1-form anomalies, and mixed 0&1-form anomalies, accordingly. There are two rather different types of mixed 0&1-form anomalies according to

$$0 \to H_2\big(BG_1, \mathbb{Z}\big) \otimes_{\mathbb{Z}} G_2 \to H_2\big(BG_1, G_2\big) \to \mathrm{Tor}_1^{\mathbb{Z}}\big(G_1^{\mathrm{ab}}, G_2\big) \to 0. \tag{4.14}$$

This short exact sequence splits, but not canonically.

Because $5 = 2 + 3 = 4 + 1$, there are two types of elementary rearrangements, namely $2 \rightleftharpoons 3$ and $4 \rightleftharpoons 1$. Decoding Ansatz (2.8) into geometry, we can draw the illustration of

the rearrangement $2 \rightleftharpoons 3$ as follows:

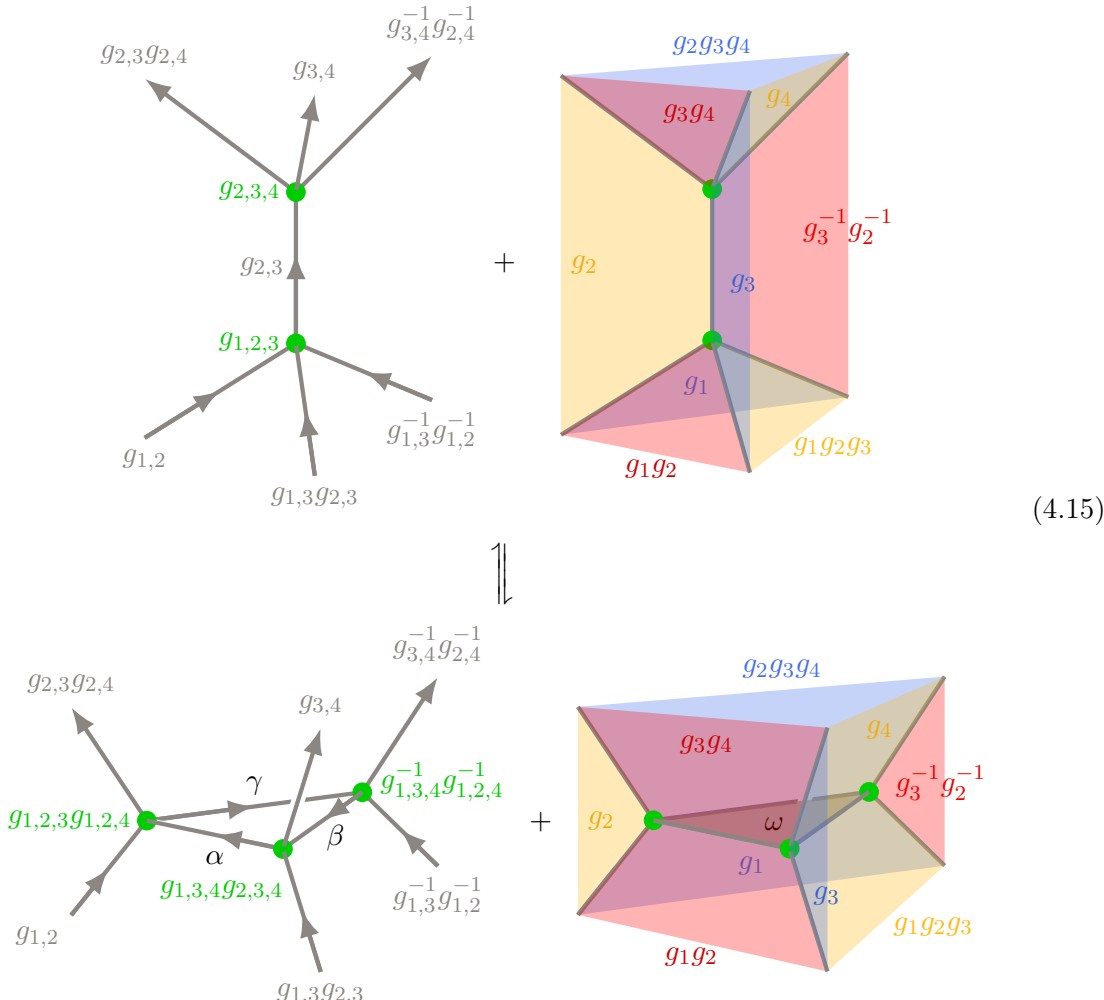

$$\tag{4.15}$$

where

$$\alpha \equiv g_{1,3}g_{1,4}g_{2,3}g_{2,4}\,, \qquad \beta \equiv g_{1,4}g_{2,4}g_{3,4}\,, \qquad \gamma \equiv g_{1,2}g_{1,3}g_{1,4}\,, \qquad \omega \equiv g_1 g_2 g_3 g_4\,, \quad (4.16)$$

and where the orientations of 2-dimensional topological operators are all (tilted) upward except for the three vertical operators, namely $g_2$, $g_3$, and $g_3^{-1}g_2^{-1}$, whose orientations obey the right-handed rule with respect to the upward direction, instead.

The first line of Eq. (4.13) gives examples of aforementioned cases, so we focus on the second line, i.e., anomalies about 0-form and 1-form symmetries. Let us start with the 1-form anomalies. It is well-known that $H^4\big(B^2 G_2, M\big)$ classifies $M$-valued quadratic forms on $G_2$ [78]. We can detect 1-form anomalies $H^4\big(B^2 G_2, U(1)\big)$ by setting only $g_{1,2}$ and $g_{3,4}$ nontrivial in the $2 \rightleftharpoons 3$ diagram (4.15). We can even further require $g_{1,2} = g_{3,4}$. Under this setting, the diagram just says that we obtain an anomalous phase when two line topological operators intersect.

From the angle of non-invertible symmetry, a finite 1-form symmetry is described by a braided fusion category, i.e., a coherent set of F-symbols and R-symbols; see [79, Chapter 8]

for a mathematical account and see [28, Appendix E] or [80] for a physical account. The infrared of gapped phases matching the anomaly can be produced by the Reshetikhin-Turaev-Witten construction. With an invertible fusion rule $G_2$, a braided fusion category reduces to an element $\in H^{\mathrm{ab}}(G_2, U(1))$, i.e., $G_2$'s Abelian cohomology [79, Sec. 8.4]. The isomorphism

$$H^4(B^2 G_2, M) = H^{\mathrm{ab}}(G_2, M) \tag{4.17}$$

is implicitly proved in [63], and the extraction of F-symbols and R-symbols from the algebraic cochains can be found in [66, Sec. 6.2]. The infrared of gapped phases matching the 1-form anomalies $H^4(B^2 G_2, U(1))$ are bosonic Abelian Chern-Simons theories, i.e., bosonic Abelian topological orders.

We can detect mixed 0&1-form anomalies $\mathrm{Hom}_{\mathbb{Z}}(H_2(BG_1, G_2), U(1))$ by setting only $g_1$, $g_2$, and $g_{3,4}$ nontrivial in the $2 \rightleftharpoons 3$ diagram (4.15). Under this setting, the diagram says that we obtain an anomalous phase when the intersection line of two surface topological operators intersects with a line topological operator. In particular, to detect the mixed 0&1-form anomalies of the type $\mathrm{Hom}_{\mathbb{Z}}(\mathrm{Tor}_1^{\mathbb{Z}}(G_1^{\mathrm{ab}}, G_2), U(1))$, it is sufficient to keep only $g_1$ and $g_{3,4}$ nontrivial. Then the diagram just says that we obtain an anomalous phase when a line topological operator fuses onto a surface topological operator. This type of mixed 0&1-form anomalies implies that the defected sector of the 1-form symmetry is charged under the 0-form symmetry, and vice versa. Tracking 0-form anomalies $H^4(BG_1, U(1))$ is more intricate in general.

From the angle of non-invertible symmetry, a finite symmetry that contains 0-form and 1-form ingredients is described by a fusion 2-category. $\mathbf{Hom}(1,1)$ of a fusion 2-category is exactly a braided fusion category. When both the 0-form and the 1-form fusion rules are invertible, the sophisticated axioms of fusion 2-categories accommodate the extensions (i.e., 2-group structures) and the anomalies. These axioms, as far as we know, are still more or less under debate; a promising solution is provided by [81].

The rearrangement $4 \rightleftharpoons 1$ is much harder to draw because of the too many overlapped 2-dimensional operators when projected onto a paper plane. Hence we merely schematically draw its 1-dimensional skeleton:

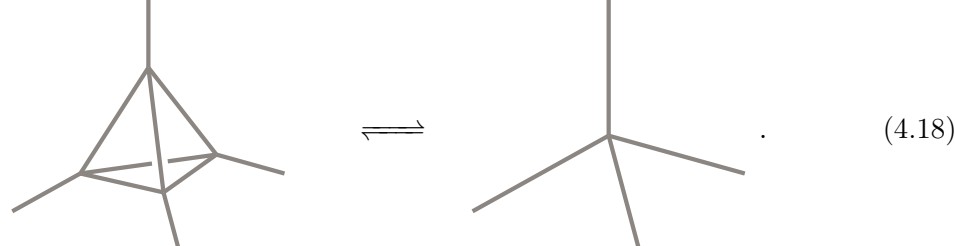

$$\tag{4.18}$$

Starting from the next dimension, i.e., 4-dimensional anomalous theories and 5-dimensional invertible field theories, there are at least 3 types of elementary rearrangement, e.g., $6 = 3 + 3 = 4 + 2 = 5 + 1$. It is also almost impossible to draw actual geometric illustrations. Nevertheless, even if we lose the control of geometric intuition, Ansatz (2.8) always enables us to write down the less intuitive but perfectly accurate mathematical formulae for the anomalies.

# 5 Outlook

Here are some prospects for future developments.

**Computation**

Cartan [82] systematically evaluated the cohomology of Eilenberg-MacLane spaces based on the bar construction [63]. Besides, a wide range of low-degree cohomology can be conveniently evaluated by Leray-Serre spectral sequences. Eilenberg and MacLane established the connection between the bar construction and the algebraic cochains implicitly based on the W construction [63]. In principle, these results enable us to extract representative algebraic cochains of a given class in $H^\bullet(G, M)$, despite the high complexity. Meanwhile, it would also be great to establish a method to compute $H^\bullet(G, M)$ from algebraic cochains and algebraic differentials.

It is also crucial to establish some methods to compute the anomaly of actual physical models. it would be useful to establish a relationship like that given in Ref. [83].

**Generalization**

We are also eager to generalize our framework to other corners in the realm of invertible symmetries, including two directions. First, we need generalized cohomologies to capture fermionic systems, discrete spacetime symmetries, gravitational anomalies, etc. Fermionic parity can be captured in a way that generalizes group supercohomologies [53–56]. Discrete spacetime symmetry can be captured by suitable $\mathbb{Z}_2$ actions on the cochains [45]. In general, we speculate that an invertible field theory can be captured by algebraic cochains as long as its partition function is nontrivial on a certain mapping torus. Nevertheless, it seems implausible that the mere algebraic cochains can capture full bordism cohomologies [e.g. the $(E_8)_1$ Chern-Simons theory, namely Kitaev's $E_8$ phase]; some decoration-type constructions like those in Ref. [84] are probably needed.

Second, we hope to describe the most general invertible symmetry. The generalization to nontrivial higher-group structures is conceptually not as hard as above. What we need to know is just how to arrange a higher-group on a simplex, as well as a convenient Ansatz to work with. Some knowledge is already known about 2-groups [8]. It is more difficult to go beyond 2-groups because the Postnikov tower is build layer by layer. The generalization to continuous symmetries may look unnecessary at the first glance because their gauge fields are better expressed by connections with curvatures instead of a web of topological operators. Nevertheless, there are often mixed anomalies between a continuous symmetry and a discrete symmetry, and webs of topological operators still make sense.

# A  Explicit differentials at low dimensions

We explicit enumerate the low-dimensional algebraic differentials defined in Def. 2.6 up to dimension $\leq 4$ based on the low-dimensional face maps enumerated in Table 1. Each column in Table 1 gives a summand in Eq. (2.25) and the contributions of adjacent columns differ by a minus sign.

| $2 \to 1$ | $\bullet = 0$ | $\bullet = 1$ | $\bullet = 2$ |
|---|---|---|---|
| $(\partial_\bullet g)_1$ | $g_2$ | $g_1 g_2$ | $g_1$ |

| $3 \to 2$ | $\bullet = 0$ | $\bullet = 1$ | $\bullet = 2$ | $\bullet = 3$ |
|---|---|---|---|---|
| $(\partial_\bullet g)_1$ | $g_2$ | $g_1 g_2$ | $g_1$ | $g_1$ |
| $(\partial_\bullet g)_2$ | $g_3$ | $g_3$ | $g_2 g_3$ | $g_2$ |
| $(\partial_\bullet g)_{1,2}$ | $g_{2,3}$ | $g_{1,3} g_{2,3}$ | $g_{1,2} g_{1,3}$ | $g_{1,2}$ |

| $4 \to 3$ | $\bullet = 0$ | $\bullet = 1$ | $\bullet = 2$ | $\bullet = 3$ | $\bullet = 4$ |
|---|---|---|---|---|---|
| $(\partial_\bullet g)_1$ | $g_2$ | $g_1 g_2$ | $g_1$ | $g_1$ | $g_1$ |
| $(\partial_\bullet g)_2$ | $g_3$ | $g_3$ | $g_2 g_3$ | $g_2$ | $g_2$ |
| $(\partial_\bullet g)_3$ | $g_4$ | $g_4$ | $g_4$ | $g_3 g_4$ | $g_3$ |
| $(\partial_\bullet g)_{1,2}$ | $g_{2,3}$ | $g_{1,3} g_{2,3}$ | $g_{1,2} g_{1,3}$ | $g_{1,2}$ | $g_{1,2}$ |
| $(\partial_\bullet g)_{1,3}$ | $g_{2,4}$ | $g_{1,4} g_{2,4}$ | $g_{1,4}$ | $g_{1,3} g_{1,4}$ | $g_{1,3}$ |
| $(\partial_\bullet g)_{2,3}$ | $g_{3,4}$ | $g_{3,4}$ | $g_{2,4} g_{3,4}$ | $g_{2,3} g_{2,4}$ | $g_{2,3}$ |
| $(\partial_\bullet g)_{1,2,3}$ | $g_{2,3,4}$ | $g_{1,3,4} g_{2,3,4}$ | $g_{1,2,4} g_{1,3,4}$ | $g_{1,2,3} g_{1,2,4}$ | $g_{1,2,3}$ |

| $5 \to 4$ | $\bullet = 0$ | $\bullet = 1$ | $\bullet = 2$ | $\bullet = 3$ | $\bullet = 4$ | $\bullet = 5$ |
|---|---|---|---|---|---|---|
| $(\partial_\bullet g)_1$ | $g_2$ | $g_1 g_2$ | $g_1$ | $g_1$ | $g_1$ | $g_1$ |
| $(\partial_\bullet g)_2$ | $g_3$ | $g_3$ | $g_2 g_3$ | $g_2$ | $g_2$ | $g_2$ |
| $(\partial_\bullet g)_3$ | $g_4$ | $g_4$ | $g_4$ | $g_3 g_4$ | $g_3$ | $g_3$ |
| $(\partial_\bullet g)_4$ | $g_5$ | $g_5$ | $g_5$ | $g_5$ | $g_4 g_5$ | $g_4$ |
| $(\partial_\bullet g)_{1,2}$ | $g_{2,3}$ | $g_{1,3} g_{2,3}$ | $g_{1,2} g_{1,3}$ | $g_{1,2}$ | $g_{1,2}$ | $g_{1,2}$ |
| $(\partial_\bullet g)_{1,3}$ | $g_{2,4}$ | $g_{1,4} g_{2,4}$ | $g_{1,4}$ | $g_{1,3} g_{1,4}$ | $g_{1,3}$ | $g_{1,3}$ |
| $(\partial_\bullet g)_{2,3}$ | $g_{3,4}$ | $g_{3,4}$ | $g_{2,4} g_{3,4}$ | $g_{2,3} g_{2,4}$ | $g_{2,3}$ | $g_{2,3}$ |
| $(\partial_\bullet g)_{1,4}$ | $g_{2,5}$ | $g_{1,5} g_{2,5}$ | $g_{1,5}$ | $g_{1,5}$ | $g_{1,4} g_{1,5}$ | $g_{1,4}$ |
| $(\partial_\bullet g)_{2,4}$ | $g_{3,5}$ | $g_{3,5}$ | $g_{2,5} g_{3,5}$ | $g_{2,5}$ | $g_{2,4} g_{2,5}$ | $g_{2,4}$ |
| $(\partial_\bullet g)_{3,4}$ | $g_{4,5}$ | $g_{4,5}$ | $g_{4,5}$ | $g_{3,5} g_{4,5}$ | $g_{3,4} g_{3,5}$ | $g_{3,4}$ |
| $(\partial_\bullet g)_{1,2,3}$ | $g_{2,3,4}$ | $g_{1,3,4} g_{2,3,4}$ | $g_{1,2,4} g_{1,3,4}$ | $g_{1,2,3} g_{1,2,4}$ | $g_{1,2,3}$ | $g_{1,2,3}$ |
| $(\partial_\bullet g)_{1,2,4}$ | $g_{2,3,5}$ | $g_{1,3,5} g_{2,3,5}$ | $g_{1,2,5} g_{1,3,5}$ | $g_{1,2,5}$ | $g_{1,2,4} g_{1,2,5}$ | $g_{1,2,4}$ |
| $(\partial_\bullet g)_{1,3,4}$ | $g_{2,4,5}$ | $g_{1,4,5} g_{2,4,5}$ | $g_{1,4,5}$ | $g_{1,3,5} g_{1,4,5}$ | $g_{1,3,4} g_{1,3,5}$ | $g_{1,3,4}$ |
| $(\partial_\bullet g)_{2,3,4}$ | $g_{3,4,5}$ | $g_{3,4,5}$ | $g_{2,4,5} g_{3,4,5}$ | $g_{2,3,5} g_{2,4,5}$ | $g_{2,3,4} g_{2,3,5}$ | $g_{2,3,4}$ |
| $(\partial_\bullet g)_{1,2,3,4}$ | $g_{2,3,4,5}$ | $g_{1,3,4,5} g_{2,3,4,5}$ | $g_{1,2,4,5} g_{1,3,4,5}$ | $g_{1,2,3,5} g_{1,2,4,5}$ | $g_{1,2,3,4} g_{1,2,3,5}$ | $g_{1,2,3,4}$ |

**Table 1**. The face maps $\partial_\bullet : G_k^{[n:k]} \mapsto G_k^{[n-1:k]}$ for small $n$.

**A.1**  $d_1 : C^1 \to C^2$

$$d_1 c_1\Big(g_1 \,\big|\, g_2 \,\big\|\, g_{1,2}\Big) \;=\; c_1\Big(g_2\Big) - c_1\Big(g_1 g_2\Big) + c_1\Big(g_1\Big).\tag{A.1}$$

**A.2**  $d_2 : C^2 \to C^3$

$$
\begin{aligned}
&d_2 c_2\Big(g_1 \,\big|\, g_2 \,\big|\, g_3 \,\big\|\, g_{1,2} \,\big|\, g_{1,3} \,\big|\, g_{2,3} \,\big\|\, g_{1,2,3}\Big)\\[4pt]
&=\; c_2\Big(g_2 \,\big|\, g_3 \,\big\|\, g_{2,3}\Big) - c_2\Big(g_1 g_2 \,\big|\, g_3 \,\big\|\, g_{1,3} g_{2,3}\Big)\\[4pt]
&\quad+ c_2\Big(g_1 \,\big|\, g_2 g_3 \,\big\|\, g_{1,2} g_{1,3}\Big) - c_2\Big(g_1 \,\big|\, g_2 \,\big\|\, g_{1,2}\Big).
\end{aligned}\tag{A.2}
$$

**A.3**  $d_3 : C^3 \to C^4$

$$
d_3 c_3\Big(g_1 \,\big|\, g_2 \,\big|\, g_3 \,\big|\, g_4 \,\big\|\, g_{1,2} \,\big|\, g_{1,3} \,\big|\, g_{2,3} \,\big|\, g_{1,4} \,\big|\, g_{2,4} \,\big|\, g_{3,4} \,\big\|\, g_{1,2,3} \,\big|\, g_{1,2,4} \,\big|\, g_{1,3,4} \,\big|\, g_{2,3,4} \,\big\|\, g_{1,2,3,4}\Big)
$$

$$
=\; c_3\Big(g_2 \,\big|\, g_3 \,\big|\, g_4 \,\big\|\, g_{2,3} \,\big|\, g_{2,4} \,\big|\, g_{3,4} \,\big\|\, g_{2,3,4}\Big)
$$

$$
-\, c_3\Big(g_1 g_2 \,\big|\, g_3 \,\big|\, g_4 \,\big\|\, g_{1,3} g_{2,3} \,\big|\, g_{1,4} g_{2,4} \,\big|\, g_{3,4} \,\big\|\, g_{1,2,4} g_{2,3,4}\Big)
$$

$$
+\, c_3\Big(g_1 \,\big|\, g_2 g_3 \,\big|\, g_4 \,\big\|\, g_{1,2} g_{1,3} \,\big|\, g_{1,4} \,\big|\, g_{2,4} g_{3,4} \,\big\|\, g_{1,2,4} g_{1,3,4}\Big)
$$

$$
-\, c_3\Big(g_1 \,\big|\, g_2 \,\big|\, g_3 g_4 \,\big\|\, g_{1,2} \,\big|\, g_{1,3} g_{1,4} \,\big|\, g_{2,3} g_{2,4} \,\big\|\, g_{1,2,3} g_{1,2,4}\Big)
$$

$$
+\, c_3\Big(g_1 \,\big|\, g_2 \,\big|\, g_3 \,\big\|\, g_{1,2} \,\big|\, g_{1,3} \,\big|\, g_{2,3} \,\big\|\, g_{1,2,3}\Big)
$$

$$\tag{A.3}$$

## A.4  $\mathrm{d}_4 : C^4 \to C^5$

$$\mathrm{d}_4 c_4 \Big( g_1 \mid g_2 \mid g_3 \mid g_4 \mid g_5 \,\Big\|\, g_{1,2} \mid g_{1,3} \mid g_{2,3} \mid g_{1,4} \mid g_{2,4} \mid g_{3,4} \mid g_{1,5} \mid g_{2,5} \mid g_{3,5} \mid g_{4,5} \Big\|$$
$$g_{1,2,3} \mid g_{1,2,4} \mid g_{1,3,4} \mid g_{2,3,4} \mid g_{1,2,5} \mid g_{1,3,5} \mid g_{1,4,5} \mid g_{2,3,5} \mid g_{2,4,5} \mid g_{3,4,5} \Big\|$$
$$g_{1,2,3,4} \mid g_{1,2,3,5} \mid g_{1,2,4,5} \mid g_{1,3,4,5} \mid g_{2,3,4,5} \,\Big\|\, g_{1,2,3,4,5} \Big)$$

$$= \; c_4 \Big( g_2 \mid g_3 \mid g_4 \mid g_5 \,\Big\|\, g_{2,3} \mid g_{2,4} \mid g_{3,4} \mid g_{2,5} \mid g_{3,5} \mid g_{4,5} \Big\|$$
$$g_{2,3,4} \mid g_{2,3,5} \mid g_{2,4,5} \mid g_{3,4,5} \,\Big\|\, g_{2,3,4,5} \Big)$$

$$- \; c_4 \Big( g_1 g_2 \mid g_3 \mid g_4 \mid g_5 \,\Big\|\, g_{1,3} g_{2,3} \mid g_{1,4} g_{2,4} \mid g_{3,4} \mid g_{1,5} g_{2,5} \mid g_{3,5} \mid g_{4,5} \Big\|$$
$$g_{1,3,4} g_{2,3,4} \mid g_{1,3,5} g_{2,3,5} \mid g_{1,4,5} g_{2,4,5} \mid g_{3,4,5} \,\Big\|\, g_{1,3,4,5} g_{2,3,4,5} \Big)$$

$$+ \; c_4 \Big( g_1 \mid g_2 g_3 \mid g_4 \mid g_5 \,\Big\|\, g_{1,2} g_{1,3} \mid g_{1,4} \mid g_{2,4} g_{3,4} \mid g_{1,5} \mid g_{2,5} g_{3,5} \mid g_{4,5} \Big\|$$
$$g_{1,2,4} g_{1,3,4} \mid g_{1,2,5} g_{1,3,5} \mid g_{1,4,5} \mid g_{2,4,5} g_{3,4,5} \,\Big\|\, g_{1,2,4,5} g_{1,3,4,5} \Big) \tag{A.4}$$

$$- \; c_4 \Big( g_1 \mid g_2 \mid g_3 g_4 \mid g_5 \,\Big\|\, g_{1,2} \mid g_{1,3} g_{1,4} \mid g_{2,3} g_{2,4} \mid g_{1,5} \mid g_{2,5} \mid g_{3,5} g_{4,5} \Big\|$$
$$g_{1,2,3} g_{1,2,4} \mid g_{1,2,5} \mid g_{1,3,5} g_{1,4,5} \mid g_{2,3,5} g_{2,4,5} \,\Big\|\, g_{1,2,3,5} g_{1,2,4,5} \Big)$$

$$+ \; c_4 \Big( g_1 \mid g_2 \mid g_3 \mid g_4 g_5 \,\Big\|\, g_{1,2} \mid g_{1,3} \mid g_{2,3} \mid g_{1,4} g_{1,5} \mid g_{2,4} g_{2,5} \mid g_{3,4} g_{3,5} \Big\|$$
$$g_{1,2,3} \mid g_{1,2,4} g_{1,2,5} \mid g_{1,3,4} g_{1,3,5} \mid g_{2,3,4} g_{2,3,5} \,\Big\|\, g_{1,2,3,4} g_{1,2,3,5} \Big)$$

$$- \; c_4 \Big( g_1 \mid g_2 \mid g_3 \mid g_4 \,\Big\|\, g_{1,2} \mid g_{1,3} \mid g_{2,3} \mid g_{1,4} \mid g_{2,4} \mid g_{3,4} \Big\|$$
$$g_{1,2,3} \mid g_{1,2,4} \mid g_{1,3,4} \mid g_{2,3,4} \,\Big\|\, g_{1,2,3,4} \Big) \, .$$

## Acknowledgments

The author is grateful to Linhao Li and Yuya Tanizaki for their numerous beneficial communications and their invaluable comments on the early draft. He thanks Aleksey Cherman for his meticulous suggestions that have vastly improved the paper's readability. He also thanks Yuji Tachikawa for sharing his encouraging opinions on the manuscript. This work was supported by Simons Foundation through the Collaboration on Confinement and QCD Strings under award number 994302.

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
