# Peer review of "Anomaly and invertible field theory with higher-form symmetry: Extended group cohomology"

_SciPost Physics_

## Round 2 · Referee Report · Anonymous (Referee 1) · 2024-7-15

Report

This paper is an initial step in generalising the algebraic approach (that is, in term of explicit cochains acting on simplices) for describing invertible field theories. The paper focuses on cases with finite higher form symmetries but no higher group structure.

There are clearly many interesting directions in which to extend this work, but it is a good first step in an interesting direction, and it contains interesting results. It is also exemplary in its clarity: the discussion is pedagogical throughout.

I recommend publication in SciPost once the (very minor) point below is addressed.

Requested changes

  1. In pg. 9 there's a discussion on gauge invariance, relating independence of the action with respect to the choice of triangulation to invariance under subdivision. But the relation between both things is not trivial, and a reference/more detailed discussion would be welcome here.

Recommendation

Ask for minor revision

---

## Round 2 · Referee Report · Anonymous (Referee 2) · 2024-8-5

Report

This paper explores the construction of an algebraic approach to the singular cohomology of $H^*(B\mathbb{G}, U(1))$, with $\mathbb{G}$ a trivial higher-group, and its application to SPT phases protected by $\mathbb{G}$. The explicit formulas constructed in the paper are useful, and it is written extremely well with many interesting follow-up directions.

Before I can recommend the paper for publication, I ask that the author strongly consider the two following suggested revisions. 1) The author does a wonderful job at pedagogically introducing topics from simplicial homotopy theory and topics related to SPTs and ’t Hooft anomalies in Euclidean field theory models where spacetime is a simplicial complex. However, It feels that the vast majority of the paper is reviewing known things, and it is often hard to distinguish between what is review and what is a new result from the author. For example, much of Sections II and III can be found in Wen’s papers arXiv:1808.09394 and arXiv:2310.08554, where the simplicial homotopy theory discussed is reviewed for nontrivial higher groups — opposed to the trivial higher groups considered here. To streamline the paper, and given that this is a physics paper with math that has appeared elsewhere in the physics literature, I strongly suggest the author reorganizes the paper to place the review of relevant math in appendices. Relatedly, often times claims throughout section II are made with no references added, and it would be helpful for the author to cite papers where the reader can go to verify the claims.

2) I think the author needs to expand Section IV to include examples to contextualize their mathematical formulae in the context of actual examples of SPTs and ’t Hooft anomalies (i.e., contextualize their results for different extended groups). Before I can recommend the paper for publication, the paper needs at least one physics example to support the mathematical theory discussed. For example, the author could contextualize their results in 1-form SPTs in 4 spacetime dimensions whose boundaries realize anomalous finite invertible 1-form symmetries captured by K matrix $U(1)$ Chern-Simons theories. These are standard 1-form SPTs, occur in oblique confined phases of $U(1)^n$ gauge theory, and can be easily generalized to have a trivial 2-group structure by enriching the boundary K matrix U(1) Chern-Simons theory by a 0-form symmetry (i.e., having the boundary be an anomalous Symmetry Enriched Topological order (SET) or an SET with symmetry fractionalization). It would also be helpful for the author to further investigate what they call ``anomaly overflow’’ instead of ignoring it. This seems interesting and new at first glance, maybe unphysical. But it looks like an interesting opportunity to test their mathematical theory in the context of physical models. Relatedly, in such examples, it would be important for the author to relate their algebraic formulation to the formulation in terms of background higher-group gauge fields that is often used when studying anomaly inflow.

Below are some minor comments/questions:

The author claims that all invertible symmetries can be described by higher groups, which are morally extensions of higher-form symmetries by lower-form symmetry, describing the dressing of junctions of lower codimension topological defects by higher codimension ones. But what about invertible symmetries for which a lower-form symmetry extends a higher-form symmetry (such an example can be found in Section 3.3 of Tachikawa’s seminal paper arXiv:1712.09542).

The author calls Kitaevs' $E_8$ state an SPT a few times, but this is not correct. It is a bosonic invertible state without any symmetry (i.e., it is the generator of the group $\mathbb{Z}$ describing 3-dimensional bosonic invertible theories).

Could the author provide references related to the following statements/parts of the paper: -"A discrete higher-group is completely determined by the homotopy type of its classifying spaces. Conversely, any topological space is a classifying space of a discrete higher-group." -"In particular, in spontaneous breaking of non-anomalous symmetry always emerges an extra symmetry that has a mixed anomaly with the original symmetry." -"This is called anomaly inflow and establishes a bijective correspondence between n-dimensional anomalies and (n + 1)-dimensional invertible field theories."

Recommendation

Ask for minor revision

---

## Editorial Decision

awaiting_resubmission